# Nanoparticle delivery of a prodrug-activating bacterial enzyme leads to anti-tumor responses

Sebastian G. Huayamares [1], Liming Lian[1], Regina Rab [2], Yuning Hou [3], Afsane Radmand[4,5], Hyejin Kim [1], Ryan Zenhausern[1], Bhagelu R. Achyut[2,3], Melissa Gilbert Ross [3], Melissa P. Lokugamage[1], David Loughrey[1], Hannah E. Peck[1], Elisa Schrader Echeverri [1], Alejandro J. Da Silva Sanchez [4,5], Aram Shajii, Andrea Li[1], Karen E. Tiegreen[1], Philip J. Santangelo[1], Eric J. Sorscher [2,3] ✉ & James E. Dahlman [1] ✉

Most cancer patients diagnosed with late-stage head and neck squamous cell carcinoma are treated with chemoradiotherapy, which can lead to toxicity. One potential alternative is tumor-limited conversion of a prodrug into its cytotoxic form. We reason this could be achieved by transient and tumor-specific expression of purine nucleoside phosphorylase (PNP), an *Escherichia coli* enzyme that converts fludarabine into 2-fluoroadenine, a potent cytotoxic drug. To efficiently express bacterial PNP in tumors, we evaluate 44 chemically distinct lipid nanoparticles (LNPs) using species-agnostic DNA barcoding in tumor-bearing mice. Our lead LNP, designated LNP intratumoral (LNP^IT), delivers mRNA that leads to PNP expression in vivo. Additionally, in tumor cells transfected with LNP^IT, we observe upregulated pathways related to RNA and protein metabolism, providing insight into the tumor cell response to LNPs in vivo. When mice are treated with LNP^IT-PNP, then subsequently given fludarabine phosphate, we observe anti-tumor responses. These data are consistent with an approach in which LNP-mRNA expression of a bacterial enzyme activates a prodrug in solid tumors.

The ability to manufacture mRNA vaccines[1,2] suggests lipid nanoparticle (LNP)-mRNA drugs could be cancer nanomedicines. This concept is substantiated by LNP-mRNA drugs encoding immunomodulatory mRNA delivered intratumorally into patients[3–5] as well as preclinical results demonstrating that intratumoral LNP-mRNA drugs can lead to anti-tumor responses. In one example, tumor growth was inhibited in a B16F10 murine melanoma model following LNP-mRNA drugs encoding immunostimulatory cytokines[6]. Other examples have demonstrated that intratumoral injection of self-replicating mRNAs,

circular mRNAs, or mRNAs encoding gene editing constructs could drive anti-tumor responses[7–9]. These data support the exploration of LNPs that efficiently deliver mRNA after intratumoral administration.

One tumor type in which intratumoral delivery could be impactful is head and neck squamous cell carcinoma (HNSCC). Head and neck cancer is the sixth most common malignancy worldwide[10], accounts for 3-4% of cancers in the United States, and is expected to increase 30% by 2030[11]. Late-stage HNSCC afflicts otherwise healthy patients and can confer chronic pain, uncontrolled bleeding, debility, and

[1]Wallace H. Coulter Department of Biomedical Engineering, Georgia Institute of Technology and Emory University School of Medicine, Atlanta, GA, USA. [2]Department of Pediatrics, Emory University, Atlanta, GA, USA. [3]Winship Cancer Institute, Emory University, Atlanta, GA, USA. [4]Petit Institute for Bioengineering and Biosciences, Georgia Institute of Technology, Atlanta, GA, USA. [5]Department of Chemical Engineering, Georgia Institute of Technology, Atlanta, GA, USA. ✉e-mail: esorscher@emory.edu; james.dahlman@emory.edu

death[10]. Patients are given surgery and/or chemoradiotherapy as a first-line treatment approach; however, this often leads to insufficient anti-tumor activity and significant toxicity[12].

Given that HNSCC is often characterized by tumors that cannot be fully resected due to their location yet can be accessed via an intra-tumoral injection, we reasoned that we could design a two-step anti-tumor therapy (Fig. 1a). In the first step, an LNP would be formulated to carry mRNA and injected intratumorally. The mRNA would encode *Escherichia coli* (*E. coli*) purine nucleoside phosphorylase (PNP), an enzyme that cleaves fludarabine to 2-fluoroadenine (F-Ade), which is cytotoxic. In the second step, we would administer fludarabine phosphate; this would be converted to fludarabine by endogenous enzymes[13], then subsequently cleaved to F-Ade in tumor cells expressing PNP[14]. Since *E. coli* PNP is not expressed in human cells, and because fludarabine is a poor substrate for human PNP analogs[15], we expected that resulting cell death would not extend beyond the tumor (i.e., where LNP-mediated PNP expression occurs). This is supported by existing clinical data. In a phase 1 clinical trial, 12 patients were treated with an adenoviral vector expressing PNP, then given fludarabine phosphate[16]. All 12 completed the study without dose-limiting toxicity, and a dose-dependent tumor response, including tumor regressions in some patients, was observed[16]. While such data are promising, adenoviral vectors can be limited by pre-existing antibodies in humans[17] that make them difficult to redose[18]. By contrast, LNP-RNA drugs are regularly readministered to patients. An LNP-mRNA-based approach would require efficient mRNA delivery after intratumoral injection.

Here, we use species-agnostic nanoparticle delivery screening (SANDS)[19] to evaluate 44 chemically distinct LNPs for intratumoral delivery in mice carrying FaDu-based tumors, a model of HNSCC[20,21]. This high-throughput in vivo LNP selection contrasts previous work optimizing LNP tumor delivery in cell culture, which can poorly predict delivery in adult animals[22]. The in vivo studies identify a lead LNP, termed LNP intratumoral (LNP[IT]), which subsequently delivers PNP-encoding mRNA to tumor models in vivo. Consistent with our hypothesis, intratumoral administration of LNP[IT]-PNP mRNA followed by fludarabine phosphate leads to anti-tumor responses in vivo without overt systemic toxicity.

## Results

### In vivo identification of an LNP for intratumoral delivery

We first evaluated whether four chemically distinct ionizable lipids that have different in vivo tropism would deliver mRNA to FaDu tumors in *NU/J* immunocompromised mice (Fig. 1b). We tested (1) DLin-MC3-DMA, which delivers siRNA to the liver in humans after an intravenous administration[23,24], (2) cKK-E12, which delivers siRNA to the liver in non-human primates (NHPs) after an intravenous administration[25], (3) SM-102, which is used in the Moderna COVID vaccine[26], and (4) C12-200, which delivers siRNA to the liver in NHPs[27] (Supplementary Table 1a). We formulated the LNPs to carry mRNA encoding anchored nanolu-ciferase (NanoLuc), injected these intratumorally at a dose of 5 μg/tumor, and measured luminescence 48 hours later (Fig. 1c). Since C12-200 had the highest bioluminescence, we performed a screen[19] using C12-200-like lipids to optimize intratumoral mRNA delivery (Fig. 1d, e). We designed 64 LNPs using stereopure isoforms of the C12-200 lipid, which were recently shown to deliver mRNA more efficiently than racemic C12-200[28], as well as the previously studied[29] polyethylene glycol (PEG)-lipid $C_{18}PEG_{2000}$. We synthesized eight lipids in total: four lipid lengths, with each chiral form (S and R) (Fig. 1f). Since cholesterol[30-32] and helper lipid structure[33-36] can affect delivery, we varied these constituents as well (Fig. 1g). Finally, to control for molar ratio-dependent effects, we formulated each potential combination with two molar ratios (Supplementary Table 1b).

We constructed all 64 LNPs to carry a DNA barcode as well as mRNA encoding a glycosylphosphatidylinositol (GPI)-anchored camelid single-variable domain on a heavy chain (VHH) antibody (anchored-VHH, aVHH)[19,37]. LNP-1, with chemical composition 1, carried aVHH mRNA and DNA barcode 1, whereas LNP-N, with chemical composition N, carried aVHH mRNA and DNA barcode N. By using a sensitive DNA barcode[38], we were able to formulate the mRNA:barcode at a ratio of 10:1. After mixing lipid and nucleic acid phases together using microfluidics[39], we characterized hydrodynamic diameter and stability of each LNP using dynamic light scattering (DLS). LNPs with a hydrodynamic diameter between 50 and 200 nm were deemed acceptable for injection. Of the 64 LNPs initially formulated, 44 met inclusion criteria and were pooled, dialyzed into PBS, and sterile filtered. As a control, we measured the hydrodynamic diameter of the pooled LNPs and found it to be within the range of diameters of LNPs constituting the pool, suggesting that the LNPs did not aggregate (Fig. 1h).

We then intratumorally administered the pooled LNP library to *NU/J* mice carrying FaDu HNSCC xenografts at a total nucleic acid dose of 6 μg/tumor (i.e., 0.13 μg/LNP, for all 44 LNPs on average). As a negative control, we added an unencapsulated DNA barcode, which is endocytosed and delivered into the cancer cells less efficiently than barcodes within LNPs[22]. Sixteen hours later, which is sufficient for cells to express the aVHH protein, we digested tumors, isolated human (i.e., human CD47[+]) cells that were functionally transfected (i.e., aVHH[+]) via fluorescence-activated cell sorting (FACS), and sequenced the CD47[+] aVHH[+] cells to identify the barcodes within them (Supplementary Fig. 1). As a final control, we measured the normalized delivery[40] of barcodes carried in LNPs and unencapsulated barcodes. As expected, unencapsulated barcodes were found less frequently than barcodes carried by LNPs (Fig. 1i).

We subsequently analyzed the barcodes to identify LNP characteristics found in both top and bottom performing LNPs (Fig. 2a, Supplementary Fig. 2). We found that cationic DC-cholesterol[41] was negatively enriched and neutral cholesterol was positively enriched; that is, the best-performing LNPs were formulated with cholesterol whereas the worst-performing LNPs were formulated with DC-cholesterol. We then calculated the normalized delivery for each LNP in the screen (Supplementary Table 2), which led us to select two lead LNPs: LNP[28] and LNP[IT]. LNP[28] and LNP[IT] were formulated with neutral cholesterol, contained the same molar ratios of four components, and formed stable LNPs with hydrodynamic diameters less than 200 nm (Fig. 2b, Supplementary Fig. 3a). When compared with the LNP composed of racemic C12-200, both LNP[28] and LNP[IT] yielded higher transfection in FaDu cells (Supplementary Fig. 3b).

We next formulated lead LNPs with anchored NanoLuc-encoding mRNA and injected these at a dose of 3 μg/tumor into bilateral FaDu tumors in *NU/J* mice. Two days later, we isolated tumors and off-target tissues and quantified NanoLuc protein expression using an in vivo imaging system (IVIS). We observed luciferase expression in the FaDu tumors (Fig. 2c, Supplementary Fig. 3c). As a control for mRNA-specific effects and to compare delivery in mouse parenchymal and human tumor cells, we formulated the lead LNPs with mRNA encoding aVHH and injected at a dose of 6 μg/tumor. Sixteen hours later, we observed LNP-mediated transfection in parenchymal and tumor (human CD47[+]) cells (Fig. 2d, Supplementary Fig. 3d, e). Consistent with the fact that *NU/J* mice lack T cells and show partial defects in B cell development due to the *Foxn1*[nu] homozygous mutation, we observed no T cells. Of the few immune cells we did observe, most were macrophages, which is consistent with previous characterizations of *NU/J* mice[42].

We then quantified delivery in a patient-derived xenograft (PDX) model. We inoculated *NOD scid gamma (NSG)* immunocompromised mice with patient-derived mixed/crude (non-clonal) HNSCC tumor cells extracted from a human lateral neck soft mass (328373-195-R-J1-PDC). We again compared LNP[28] and LNP[IT], first with mRNA encoding NanoLuc. PDX tumors injected with LNP[IT] had high bioluminescence

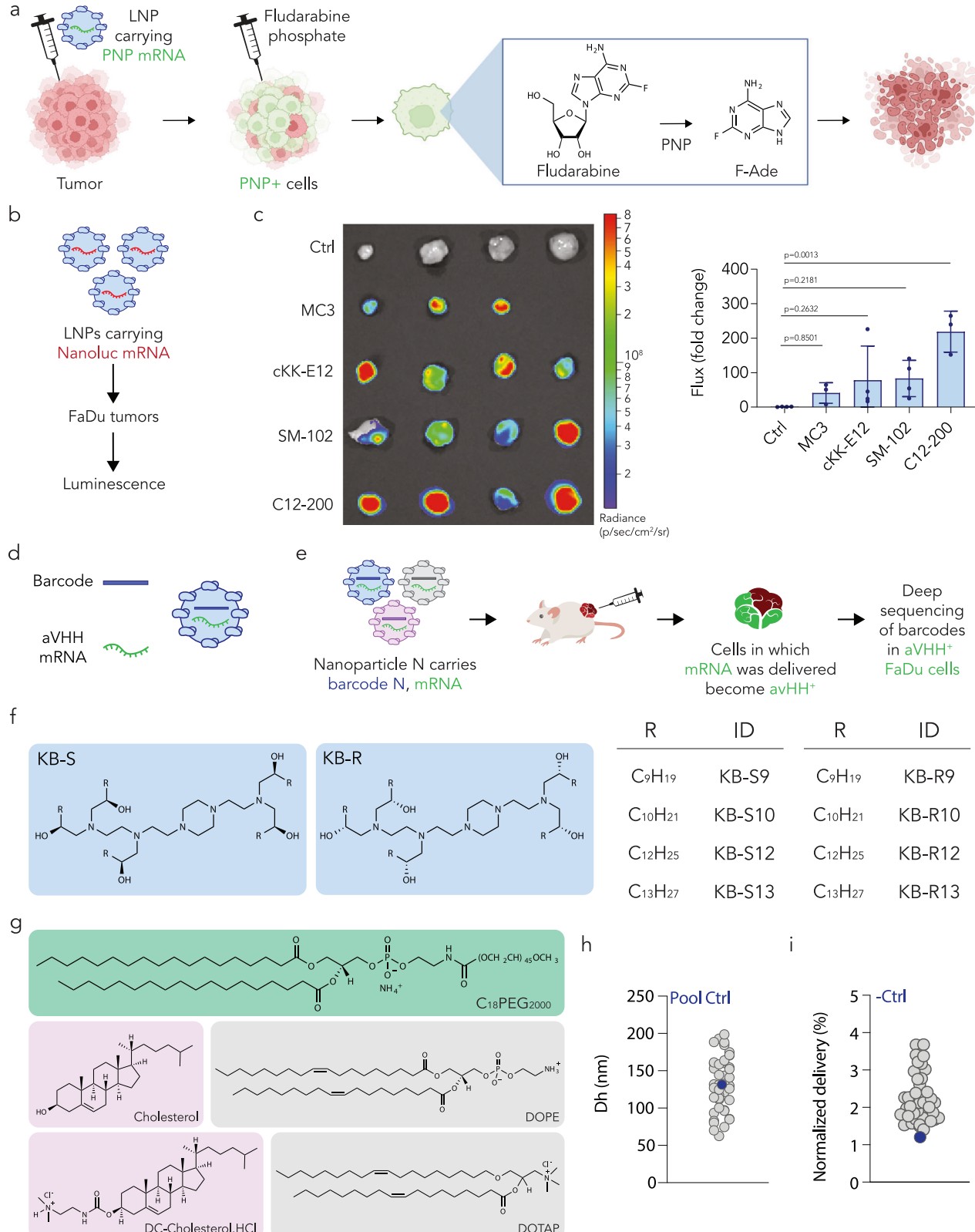

(Fig. 3a, Supplementary Fig. 4a). LNP[28] transfected PDX tumors less effectively than LNP[IT], highlighting the utility of comparing LNPs in multiple tumor models. We then repeated the experiment with aVHH; once again, LNP[28] delivered mRNA less efficiently than LNP[IT], which transfected human PDX cancer cells as well as murine cells (Fig. 3c, Supplementary Fig. 4b). The extracellular matrix and tumor micro-environment (TME) can contribute to intratumoral retention of

therapeutic agents[43,44] and may interact differently with constituents of LNPs. Differences in microenvironment may provide one potential mechanism that helps explain the gene transfer performance of LNP[28] versus LNP[IT]. Notably, *NSG* mice lack B and T cells; thus, we found fewer immune cells than in the FaDu model. To confirm the activity of LNP[IT], we increased the dose to 40 μg/tumor; this led to the expected dose-dependent increase in delivery to PDX and mouse parenchymal cells

**Fig. 1 | High-throughput DNA barcoding can be used to optimize the intratumoral (IT) delivery of mRNA to tumor cells in vivo using LNPs. a** Shown schematically, an LNP delivers *E. coli* PNP-encoding mRNA which, in combination with fludarabine phosphate, leads to cytoreduction of human HNSCC cancer cells. **b** In order to identify the most suitable ionizable lipid for intratumoral delivery, four different LNPs were formulated carrying NanoLuc reporter mRNA, intratumorally injected into FaDu xenografts, and imaged. **c** In vivo imaging system (IVIS) was used to quantify the bioluminescence resulting from the functional delivery of NanoLuc mRNA using LNPs containing four different ionizable lipids: MC3, cKK-E12, SM-102, and C12-200. Based on these results, C12-200 was chosen as the ionizable lipid for a high-throughput screen (*n* = 3–4 experimental replicates, mean +/− SD) Data analyzed by ordinary one-way ANOVA. **d** In the screen, each LNP was formulated to carry a distinct DNA barcode and aVHH reporter mRNA. **e** Human head and neck

FaDu cancer cells were inoculated to establish xenograft hindleg tumors in *NU/J* mice. The pooled LNP library was intratumorally administered to mice; 16 hours later, once the transfected anti-human CD47⁺ cells expressed aVHH, FaDu tumors were isolated, digested, and sorted for sequencing. **f** The diverse library of LNPs was formulated using various C12-200 stereopure isoforms[28]. **g** Other LNP components utilized include C18PEG2K-lipid, cholesterol or DC-cholesterol, and DOPE or DOTAP. **h** Hydrodynamic diameter of all (*N* = 44) administered LNPs (gray) as well as the pool (*N* = 1) administered (blue). **i** Normalized delivery of (*N* = 43) LNPs in tumor cells as well as the (*N* = 1) unencapsulated barcode negative control (blue). Source data are provided as a Source Data file. Figure 1a, b, d, e were created with BioRender.com released under a Creative Commons Attribution-NonCommercial-NoDerivs 4.0 International license (https://creativecommons.org/licenses/by-nc-nd/4.0/deed.en).

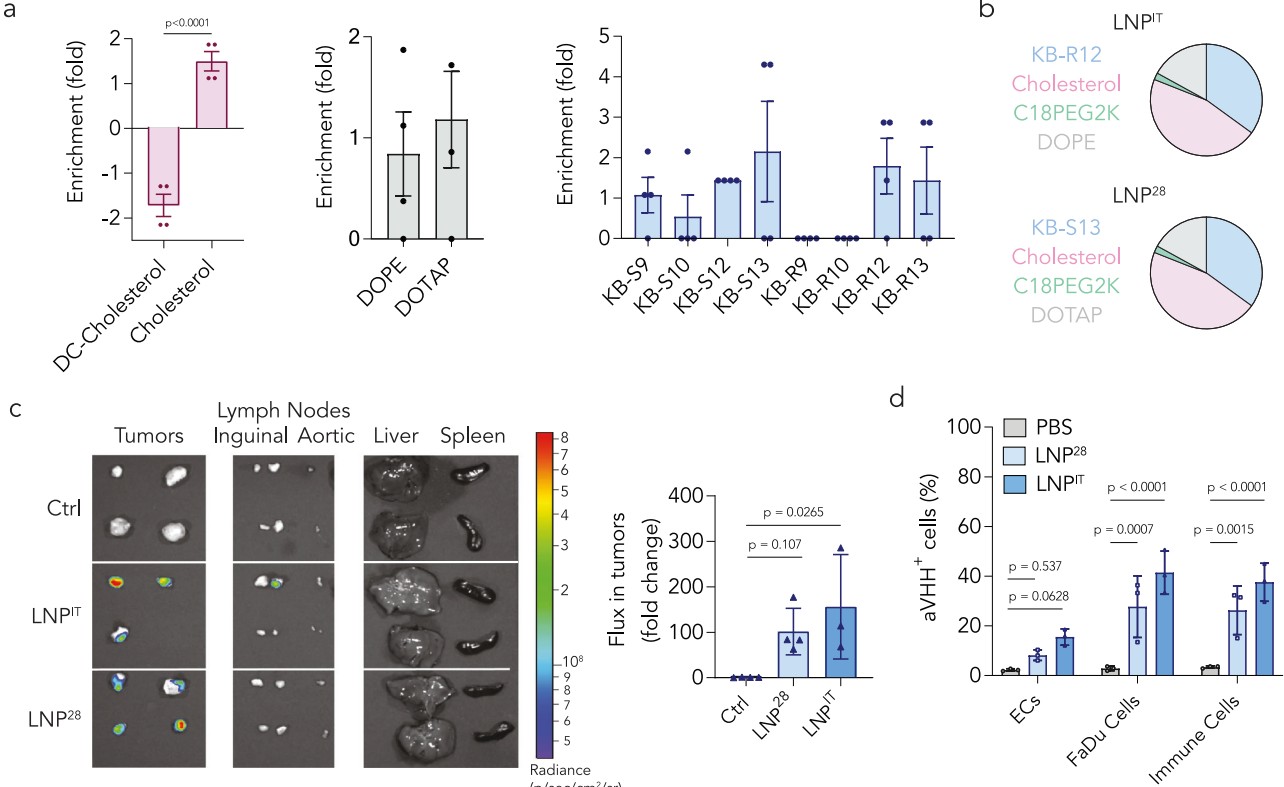

**Fig. 2 | IT LNPs that functionally deliver mRNA to human tumor cells in vivo.**
**a** Enrichment in the top 12% of LNPs screened, subdivided by the type of cholesterol, helper lipid, and stereopure ionizable lipid (*n* = 4 experimental replicates, mean +/− SD). Data analyzed by two-tailed unpaired student's *t*-test. **b** LNP²⁸ and LNPᴵᵀ were identified from the screen based on enrichment of the stereopure ionizable lipids and other components. **c** When injected intratumorally, both IT LNPs functionally delivered NanoLuc mRNA to the tumors as demonstrated by the quantification of bioluminescence via IVIS (*n* = 3-4 experimental replicates, mean

+/− SD). Data analyzed by ordinary one-way ANOVA. **d** IT LNPs functionally delivered a second reporter mRNA tested, aVHH, to CD47⁺ human head and neck FaDu cancer cells and various infiltrating immune cells in the tumors, as quantified via flow cytometry (*n* = 3 experimental replicates, mean +/− SD). Data analyzed by two-way ANOVA with Tukey post-hoc test for multiple comparisons. (ECs: endothelial cells). We found that IT LNPᴵᵀ and IT LNP²⁸ functionally delivered two different reporter mRNA molecules to FaDu tumor cell types in vivo. Source data are provided as a Source Data file.

(Fig. 3d). We did not observe off-target delivery at the 40 μg/tumor dose (Supplementary Fig. 4d).

**Single-cell LNP delivery and cell response in PDX tumors**
One limitation of many drug delivery studies, including those above, is their inability to (i) relate LNP delivery to cell heterogeneity and (ii) quantify the detailed response to LNPs. As a result, the extent to which heterogeneity and intracellular response to LNPs affect tumor delivery in vivo remains understudied. We therefore used a bespoke single-cell RNA-sequencing (scRNA-seq)-based approach[36,45] to overlay LNPᴵᵀ-mediated aVHH delivery. After intratumorally injecting mice with 40 μg/tumor of LNPᴵᵀ-mRNA or a PBS control, we mapped aVHH

protein expression onto the t-distributed stochastic neighbor embedding (t-SNE) of 2,573 PDX tumor cells mapped to the human genome (Fig. 4a). We then examined the expression levels of canonical marker genes for aggressive human HNSCC (*KRT14, KRT17, KRT6A, KRT5, KRT19, KRT8, KRT16, KRT18, KRT6B, KRT15, KRT6C, KRTCAP3, EPCAM, SFN*)[46] and found that this 12-gene signature was expressed primarily by cells in clusters 2 and 10 (Fig. 4b). Interestingly, we found different levels of LNPᴵᵀ aVHH delivery quantified via cellular indexing of transcriptomes and epitopes by sequencing[47], with the highest amount in clusters 9 and 10 (Fig. 4c, d, Supplementary Fig. 5a). The high delivery and malignancy in cluster 10 provide one early line of evidence relating malignant gene expression to LNP-based

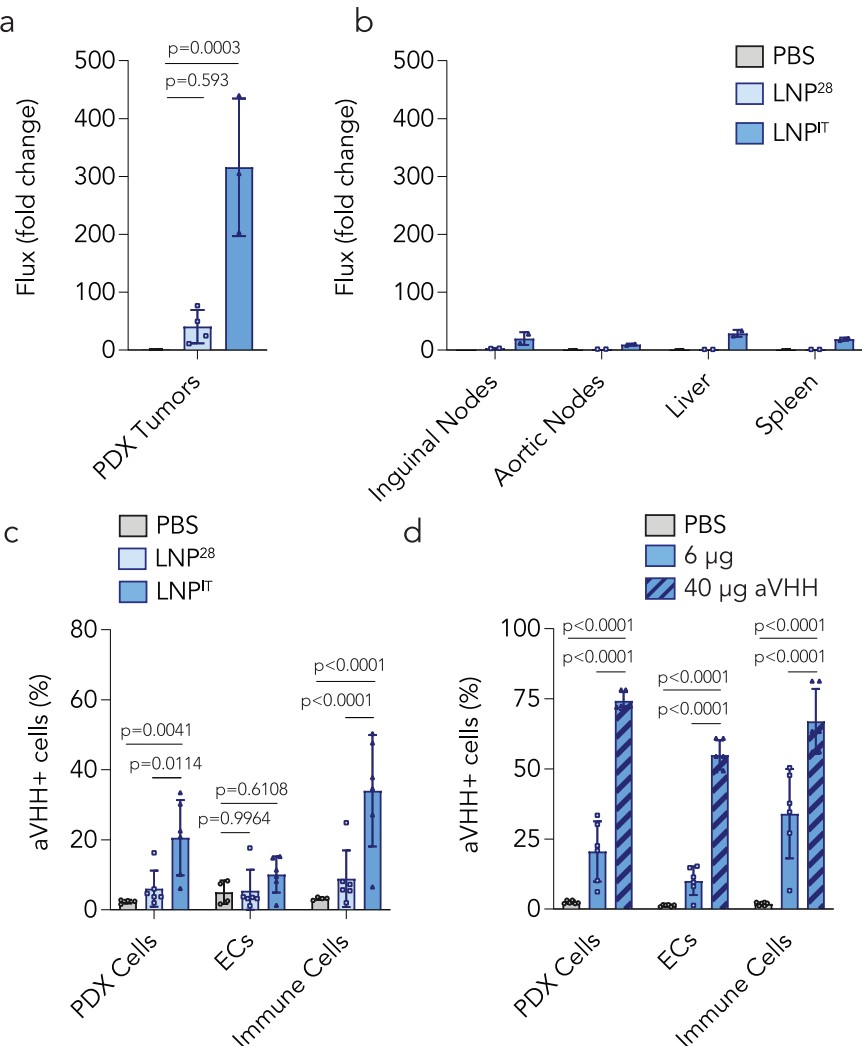

**Fig. 3 | IT LNPs can functionally deliver mRNA to tumor cell types in patient-derived xenografts in vivo.** Patient-derived xenografts (PDX) were induced in *NSG* mice, which were then injected intratumorally with the top two IT LNPs to test the delivery of two mRNAs: NanoLuc and aVHH. **a** Bioluminescence, quantified via IVIS, showed that LNP[IT] achieved better functional delivery of NanoLuc in PDX tumors than LNP[28] (*n* = 3 experimental replicates, mean +/− SD). Data analyzed by one-way ANOVA. **b** Off-target bioluminescence is presented from (*N* = 2) mice. **c** LNP[IT] functionally delivered aVHH to 20% of PDX tumor cells as well as multiple different

immune cell types (*n* = 6 experimental replicates, mean +/− SD). (ECs: endothelial cells). Data analyzed by two-way ANOVA with Tukey post-hoc test for multiple comparisons. **d** LNP[IT] was spin-concentrated and injected at 40 μg of aVHH mRNA/tumor. This significantly increased transfection levels of PDX cells to 75%, ECs to 55%, and immune cells to 67% (*n* = 6 experimental replicates, mean +/− SD). Data analyzed via two-way ANOVA with Tukey post-hoc test for multiple comparisons. Source data are provided as a Source Data file.

transfection. Although significant future work is required to validate this potential relationship, it is important to note this finding would have been difficult to observe using traditional delivery readouts.

We noted a number of variably up- or down-regulated genes in each cluster (Supplementary Fig. 5b). For example, we found clusters 2, 5, 8, and 10 had the highest gene expression of HNSCC stromal cell gene markers (*ALDH1A1, BCL11B, BMI1, CD44*)[48] (Supplementary Fig. 5c). We also found CD47 expressed ubiquitously across all clusters (Supplementary Fig. 5d). When evaluating *SLA2*, a prognostic marker in HNSCC that correlates with immune cell infiltration of the TME[49], we observed no expression (Supplementary Fig. 5e). This is consistent with low immune cell infiltration in an immunocompromised murine model.

We then studied the transcriptomic changes in transfected cancer cells by analyzing differential expression. To control for potential bystander effects in the microenvironment (compared to PBS-treated mice) (Supplementary Fig. 6a), we compared cells that were targeted

by LNPs (i.e., aVHH[+] cells) to cells that were not (aVHH[-] cells) from the same tumors. We found that 928 genes were significantly (*p* < 0.05) upregulated, including human HNSCC genes such as MYH9[50] and PRDX5[51], and 17 were downregulated in aVHH[+] cells compared to aVHH[-]. When the top significantly upregulated genes (*p* < 0.001) from aVHH[+] cells transfected by LNP[IT] (Fig. 4e) were analyzed using the ReactomeGSA database[52], we found 28 significantly enriched protein metabolic pathways (R-HSA-392499.10). Interestingly, all 28 of these pathways were related to cellular RNA or protein management. Of these, 14 were directly associated with mRNA translation into protein (R-HSA-72766) and stemmed directly from the core of the metabolic pathway tree (Biological Process GO:0019538), consistent with previously reported transcriptional responses to mRNA-carrying LNPs[36] (Fig. 4f, g, Supplementary Fig. 6b). Some of the most upregulated genes found in the PDX model in vivo were also explored in a different PDX model (PDX2) transfected with LNP[IT] (Supplementary Fig. 6c). These data provide an early line of evidence that HNSCC cancer cells

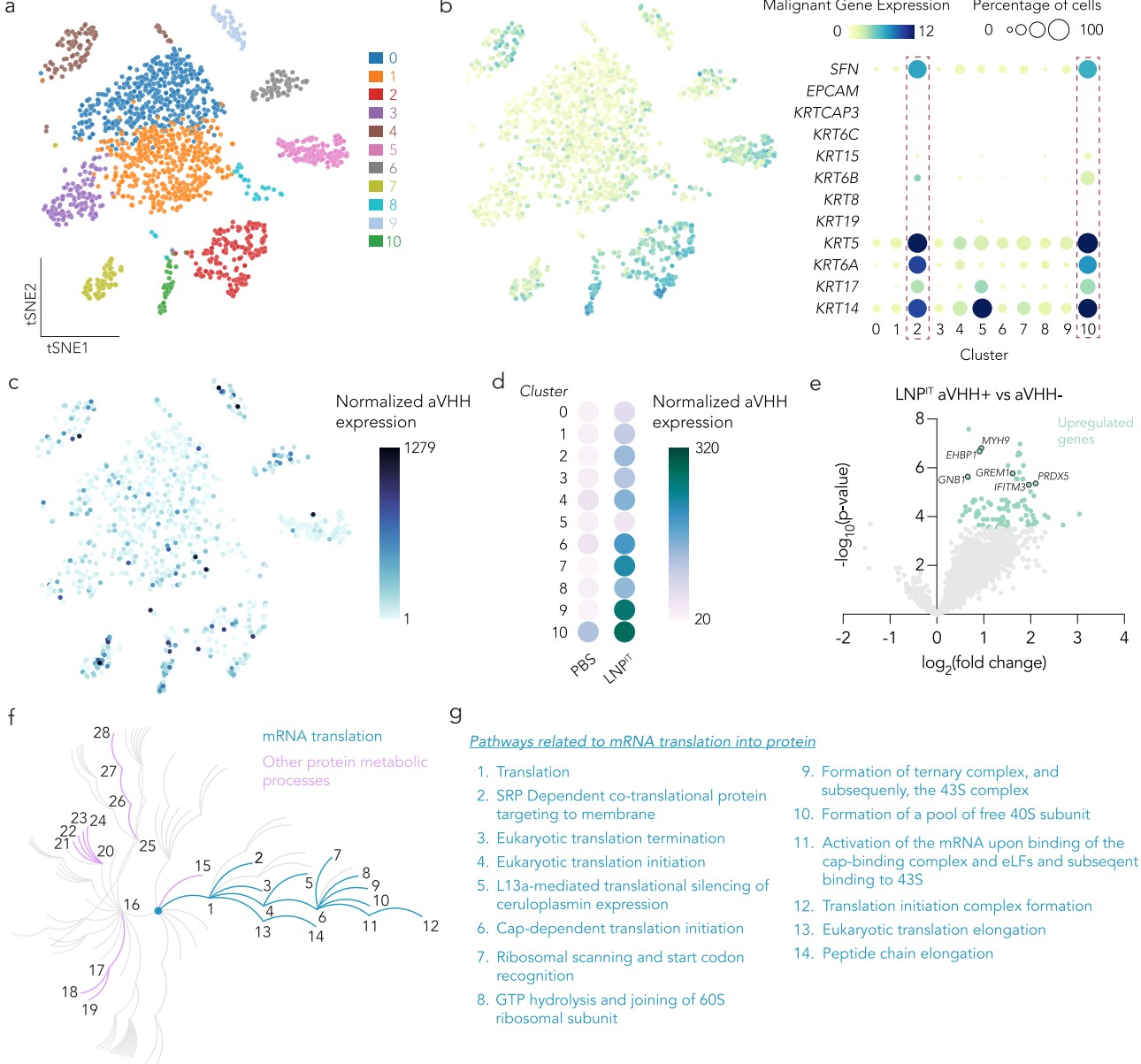

**Fig. 4 | LNP^IT transfected malignant cell types of HNSCC tumors and upregulated RNA and protein metabolic pathways. a** When mapped to the human genome, PDX tumor cells grouped into 11 different clusters using t-distributed stochastic neighbor embedding (t-SNE). **b** Cells in clusters 2 and 10 had the highest expression of 12 genes established via scRNA-seq as a prognostic malignant HNSCC cell gene signature in patients with HNSCC[46]. **c, d** LNP^IT transfected aVHH differently within each PDX HNSCC cluster. **e** aVHH⁺ cells transfected with LNP^IT compared to aVHH⁻ cells (top 80 upregulated genes, *p* < 0.001). **f** Reactome pathway analysis based on most significantly upregulated genes by LNP^IT (*p* < 0.01), showing upregulated pathways associated with the metabolism of proteins (Reactome ID: R-HSA-392499.10). **g** Twenty-eight pathways were upregulated by LNP^IT; half of these are associated with mRNA translation into protein. Source data are provided as a Source Data file.

respond to LNPs carrying mRNA in part by altering genes related to the manufacture and processing of RNA and protein.

## LNP^IT-PNP followed by fludarabine yield anti-tumor responses

The screening data, confirmation studies, and scRNA-seq results led us to conclude that LNP^IT transfected HNSCC cells in mice. We therefore used LNP^IT to test the hypothesis that LNP-PNP followed by fludarabine phosphate could have anti-tumor effects. We synthesized a chemically modified mRNA encoding *E. coli* PNP, formulated it within LNP^IT, and injected intratumorally at a dose of 6 μg/tumor. Since there is not an extensively validated antibody for PNP, we quantified tumor PNP enzymatic activity using high-performance liquid chromatography (HPLC)[15,53]. PNP activity was highest six to 24 hours after injection and

decreased at 48 hours (Fig. 5a, b), with no off-target PNP activity in the liver or spleen (Supplementary Fig. 7a, b). When we increased the PNP mRNA dose to 40 μg mRNA/tumor, we observed a dose-dependent increase in PNP activity (Fig. 5c).

We then performed a therapeutic study in *NSG* mice. After inoculating animals with $10^7$ PDX cells, we monitored tumor growth until the malignant masses reached volumes between 150 and 250 mm³. In group one, mice were then injected with LNP^IT-mRNA, then 24 hours later with fludarabine phosphate; in group two, mice were injected with LNP^IT-mRNA, then a DMSO control; in group three, a PBS control followed by fludarabine phosphate; and in group four, PBS then DMSO (Fig. 5d). Mice from group one had significantly smaller tumor volumes than each of the three control groups after treatment

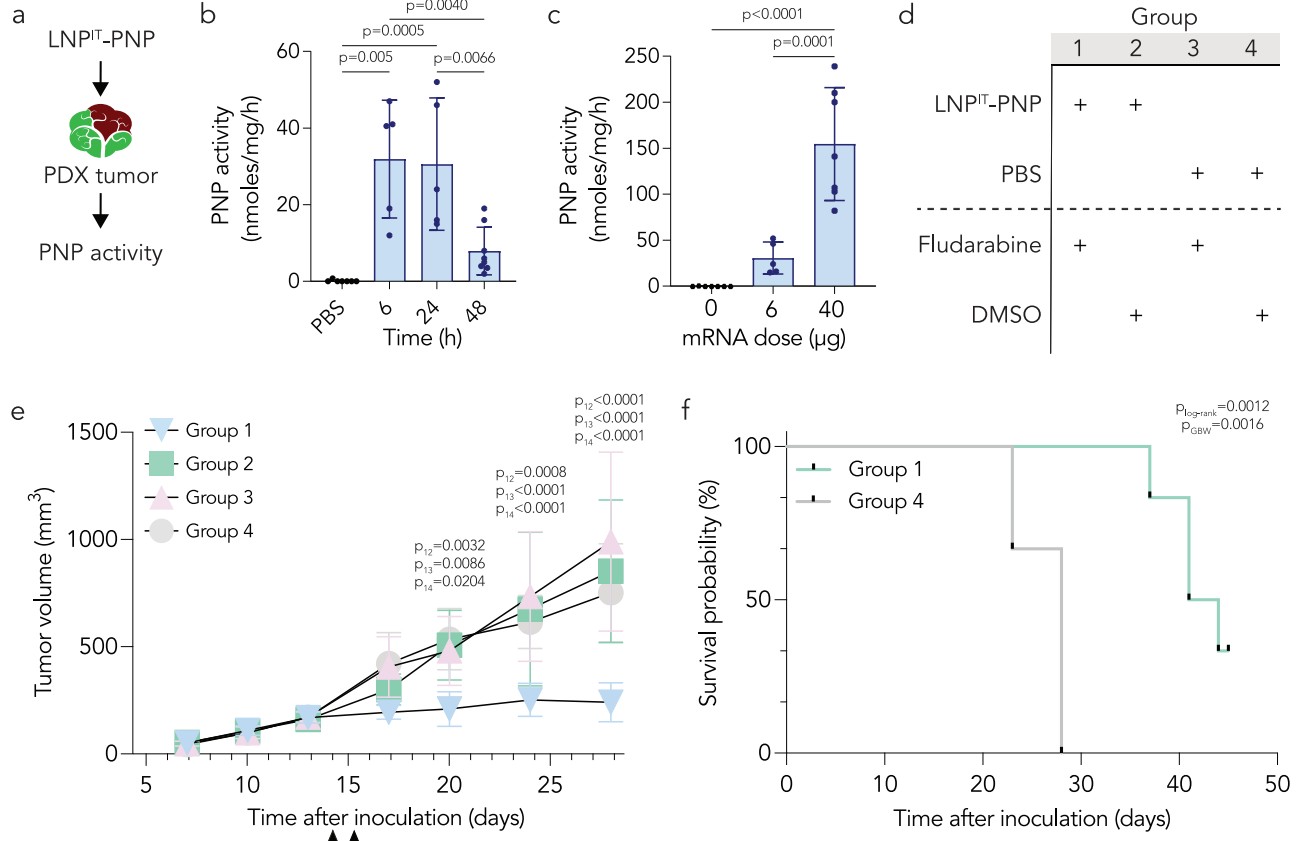

**Fig. 5 | LNP$^{IT}$ carrying PNP mRNA, combined with fludarabine phosphate, induces tumor suppression of HNSCC patient-derived xenografts. a** *E. coli* PNP transfection by LNP$^{IT}$ was quantified in PDX tumors via an HPLC-based PNP enzymatic assay. **b** PNP activity was measured in PDX tumors harvested at different timepoints to determine the pharmacokinetics of PNP mRNA expression ($n = 5$–8 experimental replicates, mean +/− SD). Data analyzed by two-way ANOVA with Tukey post-hoc test for multiple comparisons. **c** Dose response of the LNP$^{IT}$-PNP when concentrated and administered at 40 μg per PDX tumor ($n = 5$–7 experimental replicates, mean +/− SD). Data analyzed by ordinary one-way ANOVA. **d** On day 14 after tumor inoculation, LNP$^{IT}$-PNP (or PBS for control groups) was injected at 20 μg/tumor in the morning and in the afternoon (40 μg total). The next day, fludarabine phosphate (or DMSO vehicle control) was injected in the morning and

afternoon, and tumor volume measured twice a week. **e** Tumor volume was diminished for the group treated with LNP$^{IT}$-PNP and fludarabine, compared to the control groups ($n = 6$ experimental replicates, mean +/− SD). Data analyzed by two-way ANOVA with Tukey post-hoc test for multiple comparisons ($p_{12}$ for group 1 vs 2, $p_{13}$ for group 1 vs 3, $p_{14}$ for group 1 vs 4). **f** Higher probability of survival was observed in mice ($N = 6$) treated with the LNP$^{IT}$-PNP and fludarabine combination therapy ($p_{log-rank}$ calculated with the log-rank/Mantel-Cox test, $p_{GBW}$ calculated with the Gehan-Breslow-Wilcoxon test). Source data are provided as a Source Data file. Figure 5a was created with BioRender.com released under a Creative Commons Attribution-NonCommercial-NoDerivs 4.0 International license (https://creativecommons.org/licenses/by-nc-nd/4.0/deed.en).

(Fig. 5e, Supplementary Fig. 8a), and anti-tumor responses were observed as early as one day after completing treatment (Supplementary Fig. 8b). We did not observe significant weight loss in group one relative to other groups, which provides an early line of evidence suggesting the intervention was tolerated (Supplementary Fig. 8c). We then repeated the experiment with groups one and four, this time using a clinical scoring system incorporating tumor size, tumor ulceration, body condition, and mobility (Supplementary Fig. 9). Treated mice survived longer than mice treated with controls (Fig. 5f). These data are consistent with the hypothesis that LNP$^{IT}$-mRNA treatment followed by fludarabine phosphate can lead to anti-tumor responses.

Since HNSCC tumors are heterogenous, we tested therapeutic effect in two additional in vivo models: FaDu tumors in immunocompromised *NU/J* mice (Fig. 6a) and MOC1 murine oral cancers (syngeneic HNSCC) in immunocompetent C57BL/6 mice (Fig. 6b). Tumors treated with the combination therapy were significantly smaller than those in the control groups. Given that MOC1 tumors are characterized by increased MHC-I expression and CD8+ T cell infiltration into the tumor microenvironment[54] while FaDu tumors were grown in *NU/J* mice with T cell deficiency[55], these data provide

additional evidence that the approach is active in a number of different host immunologic contexts.

To evaluate therapeutic response in distinct human cancer cells, we also performed an in vitro cell killing assay (Fig. 6c). We plated six different HNSCC cell lines: human FaDu, murine MOC1, and four HNSCC patient-derived xenograft lines, termed PDX1, PDX2, PDX3, and PDX4 (Supplementary Fig. 10a). We treated cells with LNP$^{IT}$-PNP and MeP-dR (a nucleoside analogue of fludarabine that has served as a prototype for in vitro testing). We showed the expected PNP and nucleoside-dependent cell killing and also observed time-dependent conversion from prodrug to oncolytic drug (Supplementary Fig. 10b). The negative controls, LNP-PNP$^{IT}$ alone, MeP-dR alone, and no treatment, behaved as expected. In contrast, cells treated with the positive control MeP (the toxic base released following MeP-dR hydrolysis by PNP) led to the expected ablation of cells in culture.

## Discussion

LNPs delivering RNA drugs to the liver lead to clinical responses in patients[56–58]. These data suggest the potential impact of LNP-RNA drugs that target non-hepatic tissues[59]. Yet, evidence also indicates that an LNP optimized for one route of administration is not optimized

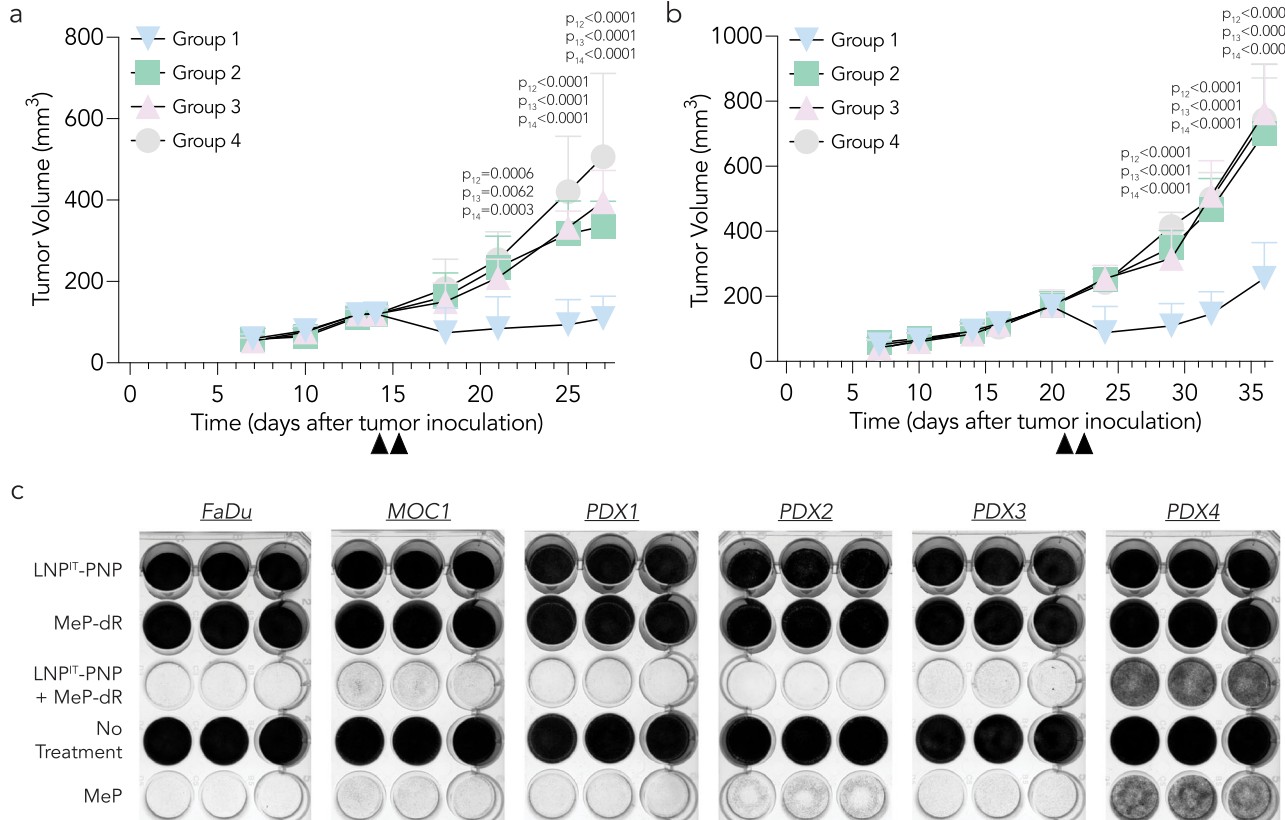

**Fig. 6 | LNP^IT-PNP and fludarabine phosphate combination therapy has antitumor effects validated across various HNSCC preclinical models.** Tumor growth studies in two additional in vivo HNSCC models: **a** FaDu, human HNSCC xenografts inoculated in immunocompromised *NU/J* mice (*n* = 4−5 experimental replicates, mean +/− SD; $p_{12}$ for group 1 vs 2, $p_{13}$ for group 1 vs 3, $p_{14}$ for group 1 vs 4), and (**b**), MOC1, syngeneic HNSCC murine tumors inoculated in immunocompetent C57BL/6 mice (*n* = 5−6 experimental replicates, mean +/− SD; $p_{12}$ for group 1 vs 2, $p_{13}$ for group 1 vs 3, $p_{14}$ for group 1 vs 4). Tumor volume was compared to the control groups using a two-way ANOVA with Tukey post-hoc test for multiple comparisons. **c** An in vitro assay to assess the anti-tumor cell effects of combination therapy in the syngeneic MOC1 murine line, FaDu human tumor cells, and four HNSCC patient-derived xenograft lines (PDX1, PDX2, PDX3, and PDX4). MeP-dR is an analogue of fludarabine phosphate (prodrug), used as a prototype compound for showing PNP activity. MeP is the toxic cleavage product of MeP-dR following PNP treatment and serves as a positive control (*n* = 3 experimental replicates consisting of wells seeded with the respective cell line). Source data are provided as a Source Data file.

for another[60], highlighting the utility of studying how LNPs behave after an intratumoral injection. Here, we used species-agnostic DNA barcoding to test 44 LNPs in tumor-bearing mice. Our results provide early evidence that LNPs formulated with neutral cholesterol may outperform LNPs formulated with cationic cholesterol for intratumoral delivery. This finding is consistent with the hypothesis that negatively charged microenvironments[61], including certain tumor microenvironments[62,63], can sequester positively charged drugs. Our analysis also suggests a second lesson: future screens should take place in multiple tumor models. Here, we evaluated all 44 LNPs in FaDu tumors, leading us to discovery of LNP[28] and LNP^IT. When the two lead LNPs were tested individually, they both delivered mRNA in FaDu tumors as predicted. Yet, when the LNPs were tested in PDX tumors, only LNP^IT efficiently delivered mRNA. This second lesson is timely; in the past, testing several dozen LNPs in multiple in vivo tumor models would have required hundreds of animals. DNA barcoding makes this feasible with far fewer. When stereopure C12-200 isoforms were previously screened intravenously, the S-isomers yielded higher expression[28], while the most effective intratumoral LNP candidate here was composed of an R-isomer. This highlights the importance of screening via the intended route of administration when selecting LNP candidates for RNA therapeutics[64].

Complementing the lessons learned as a result of barcode sequencing were the observations made by analyzing LNP-transfected tumors with scRNA-seq. We found it interesting that an unbiased approach evaluating cellular processes identified

28 significant pathways−all of which were related to RNA or protein. These data are consistent with the hypothesis that cancer cells respond to LNPs carrying mRNA by changing the ways in which mRNA and protein are processed. We foresee future work focused on understanding how these processes may be exploited to further improve LNP delivery.

It is also important to acknowledge the limitations of the study. Our applications of human-derived tumor models required testing mice with deficient immune systems. One question that will be answered in future work is whether LNP delivery occurs within other intratumoral immune cell subtypes in mice with intact immune systems and spontaneous tumors. If LNP^IT does transfect immune cells, the delivery system could be considered for immunostimulatory mRNA approaches[7], such as those that have been tested in patients[3,4,65]. A second limitation of the work involves delivery readouts; it is unclear how well delivery in mice predicts delivery in NHPs[19]. NHP studies will be particularly important to understand on- and off-target delivery and subsequent toxicity. In our experiments, we did not observe substantial off-target delivery, which likely reduced the chance of systemic toxicity. An additional source of toxicity is F-Ade released from tumor parenchyma. However, previous data suggest that escaped F-Ade would be diluted and metabolized via systemic xanthine oxidase[15,16,66] to non-toxic concentrations. Finally, the expression of PNP, which is a bacterial protein, could elicit an unexpected immune response. While this will require further study, we did not find any evidence suggesting a broad, undesired immune response in our current experiments.

Despite these limitations and need for future work, our results are consistent with the hypothesis that LNPs can deliver a bacterial enzyme that activates a prodrug and leads to anti-tumor responsiveness. We believe the data support additional preclinical work focused on evaluating this anti-cancer approach.

## Methods

All experiments and research performed comply with all relevant ethical regulations in accordance with Emory University's IACUC.

### C12-200 synthesis

Stereopure C12-200 ionizable lipids[28] were formulated by adding amine (0.2 mmol) and chiral epoxide (1.4 mmol, 7equiv) to a 10 mL reaction vial with magnetic stirring. The vial was sealed, heated to 80 °C, and stirred for 48 hours. The crude mixture was then purified by column chromatography using a silica gel (300–400 mesh) and eluting with a DCM:MeOH:NH$_4$OH (90:9:1) mixture, affording desired compounds as colorless oils in 53–62 yields. Thin layer chromatography (TLC) was carried out using precoated silica Gel GF plates and visualized using KMnO4 stains. $^1$H and $^{13}$C NMR spectra were recorded on a Bruker AVANCE 400 (400 MHz), and a Bruker AVANCE 600 (600 MHz) spectrometer at 25 °C. All $^1$H Chemical shifts (in ppm) were assigned according to CDCl$_3$ ($\delta$ = 7.24 ppm), and all $^{13}$C NMR was calibrated with CDCl$_3$ ($\delta$ = 77.00 ppm). Coupling constants (J) are reported in hertz (Hz). High-resolution mass spectra (HRMS) were recorded on LC/MS (Agilent Technologies 1260 Infinity II/6120 Quadrupole) and a time-of-flight mass spectrometer by electrospray ionization. A general reaction mechanism can be found in Supplementary Fig. 11. NMR/mass spec data for the ionizable lipids can be found in Supplementary Fig. 12.

### Messenger RNA (mRNA) synthesis

E. coli PNP was designed based on ORF sequences. UTRs consisting of a 5′ untranslated region (UTR) from Kozak sequence and a 3′ UTR from mouse alpha-globin (GenBank accession no. NM_001083955) similar to the constructs previously described[67,68] were used. The plasmids were linearized with Not-I HF (New England Biolabs) overnight at 37 °C. Linearized templates were purified by ammonium acetate (Thermo Fisher Scientific) precipitation and resuspended with nuclease-free water. In vitro transcription was performed overnight at 37 °C using the HiScribe T7 Kit (NEB) following the manufacturer's instructions (N1-methyl-pseudouridine modified). The resulting RNA was treated with DNase I (Aldevron) for 30 min to remove the template and purified using lithium chloride precipitation (Thermo Fisher Scientific). The RNA was heat denatured at 65 °C for 10 min before capping (via Cap-1 structure) using guanylyl transferase and 2′-O-methyltransferase (Aldevron). mRNA was then purified by lithium chloride precipitation, treated with alkaline phosphatase (NEB), and purified again. mRNA concentration was measured using a NanoDrop protocol. mRNA stock concentrations were ~4 mg/ml. Purified mRNA products were analyzed by gel electrophoresis to ensure purity. Messenger RNA encoding anchored nanoluciferase or anchored VHH (aVHH)[19,69] was synthesized such that GPI-anchored VHH and NanoLuc sequences were ordered as a DNA gBlock from Integrated DNA Technologies (IDT) with the same 5′ and 3′ UTRs used for PNP mRNA. Purification of linearized templates, in vitro transcription, DNase I treatment, heat denaturation, and downstream purification with lithium chloride precipitation and NEB were all performed exactly as detailed above for PNP mRNA.

### Nanoparticle formulation

Nanoparticles for in vivo screening were formulated using a microfluidic device[39] at a flow rate ratio of 3:1 of nucleic acid: lipid phases. Larger batches of LNP$^{IT}$ with PNP mRNA for preclinical tumor regression studies were formulated using the NanoAssemblr Ignite (Precision Nanosystems). DNA barcodes and/or mRNA were diluted in 10 mM citrate buffer (Teknova). DNA barcodes were purchased from IDT.

PEGs, cholesterols, and helper lipids were diluted in 100% ethanol and purchased from Avanti Lipids. Citrate and ethanol phases were combined in a microfluidic device or the NanoAssemblr Ignite using glass (Hamilton Company) or plastic (BD) syringes, respectively, at a flow rate ratio of 3:1.

### Nanoparticle characterization

The diameter and polydispersity of all LNPs were assessed via dynamic light scattering (DLS) (DynaPro Plate Reader III, Wyatt). Four microliters of LNPs were diluted in 96 microliters of sterile 1X PBS. LNPs were sterile purified using a 0.22 μm filter and injected only if three criteria were met: 20 nm <diameter <200 nm, and correlation function with 1 inflection point (monodisperse distribution). For screens, particles that met these cut-offs were pooled. Particles were dialyzed in 20 kD dialysis cassettes (Thermo Scientific). NanoDrop (Thermo Scientific) was used to assess nucleic acid concentration.

### Encapsulation assay

Encapsulation was measured according to the Precision NanoSystems RiboGreen assay protocol. In duplicates, 50 μL of 6 ng/μL LNP (diluted in TE) was added to 50 μL of 1X TE (Thermo Fisher) or 50 μL of a solution containing a 1:50 dilution of Triton X-100 (Sigma Aldrich). After 10 minutes of incubation at 37 °C, 100 μL of 1:100 of RiboGreen reagent (Thermo Fisher) was added to each well. The fluorescence was quantified using a plate reader (BioTek Synergy H4 Hybrid) at an excitation wavelength of 485 nm and an emission wavelength of 528 nm.

### Zeta potential

A Malvern Zetasizer Nano Z was used to measure the zeta potential of LNPs. Eight hundred microliters of the nanoparticles were loaded into a Malvern disposable folded capillary cell and software run under the following settings: material refractive index of 1.4, absorbance of 0.01, dispersant viscosity of 0.882cp, refractive index of 1.33, and dielectric constant of 79.

### TNS assay

A stock solution of 10 mM HEPES (Sigma Aldrich), 10 mM MES (Sigma Aldrich), 10 mM sodium acetate (Sigma Aldrich), and 140 mM sodium chloride (Sigma Aldrich) was prepared and adjusted with hydrogen chloride and sodium hydroxide to the following pH values: 3, 4, 5, 6, 7, 8, 9, 10. Using four replicates for each pH, 140 μL pH-adjusted buffer was added to a 96-well plate, followed by treatment with 5 μL of 2-(p-toluidino)-naphthalene-6-sulfonic acid (60 μg/mL). Five microliters of LNP were next added to each well, followed by incubation for 5 minutes with constant shaking at 300 rpm. Fluorescence absorbance was quantified with excitation wavelength of 325 nm and an emission wavelength of 435 nm using a plate reader (BioTek Synergy H4 Hybrid).

### Animal experiments

All animal experiments were performed in accordance with Emory University's IACUC. All animals were housed in the Emory University Winship Cancer Center Animal Facility. NU/J mice (The Jackson Laboratory, stock #002019, 6-8 weeks, female), NOD.Cg-Prkda$^{scid}$, IL2rg$^{tm1Wjl}$/SzJ (NSG) mice (The Jackson Laboratory, stock #005557, 6-8 weeks, female), and C57BL/6J mice (The Jackson Laboratory, stock #000664, 6-8 weeks, female) were used throughout this article. At least $N$ = 2−6 female mice/group were used for all studies, unless noted otherwise. NSG and NU/J mice were housed in a microisolator on static racks in animal rooms maintained at 22 ± 2 °C, and 40−50% humidity, with 12-hour light/dark cycle. Both NSG and NU/J mice were fed with sterile Rodent Lab Diet (Laboratory Rodent Diet no. 5R53) and sterile water. The C57BL/6J mice were housed in a microisolator in animal rooms maintained at 22 ± 2 °C and 40-50% humidity, with a 12-hour

light/dark cycle. C57BL/6J mice were fed with Rodent Lab Diet (Laboratory Rodent Diet no. 5053).

## Ethics statement

All animal experiments were performed in accordance with Emory University's IACUC. In all cases, the animal was euthanized when the maximal tumor diameter reached 20 mm (TBS of 3, see Tumor Volume Measurement and Clinical Scoring section below). PDX models were obtained from the NCI Patient-Derived Models Repository (PDMR; https://pdmr.cancer.gov; NCI-Frederick, Frederick National Laboratory for Cancer Research, Frederick, MD). Specimens were collected under NCI-sponsored tissue procurement protocols with institutional review board approval; investigators obtained written informed consent from each participant for the use of their delinked specimens to genetically characterize and generate patient-derived models and to make these models available to researchers along with limited clinical information.

## FaDu, PDX, and MOC1 tumor inoculation

FaDu head and neck squamous cell carcinoma cells (ATCC, catalog #HTB-43) were subcutaneously injected into *NU/J* mice to induce bilateral xenograft hindleg tumors ($1 \times 10^6$ cells per unilateral inoculation). Tumor volumes were monitored twice a week starting on day 7, once FaDu masses reached between 150 and 250 mm$^3$ in size. Patient-derived xenograft (PDX) cells were obtained as mixed/crude (non-clonal) tumor samples (e.g., 328373-195-R-J1-PDC) extracted from a lateral neck mass according to protocols at the National Cancer Institute (NCI) patient-derived models repository (PDMR) (https://pdmr.cancer.gov/). Cancer models from PDMR undergo rigorous quality control profiling and exhibit suitable doubling time and carcinogenic pathology for assessing antitumor activity in preclinical studies with reliable fidelity. *NSG* mice were injected subcutaneously with $1 \times 10^7$ PDX cells, as recommended by PDMR/NCI to model an aggressive form of HNSCC. The additional 3 PDX lines tested later (PDX2, PDX3, PDX4; Supplementary Fig. 10a) were also obtained from the NCI PDMR. MOC1 cells (Kerafast, catalog #EWL001-FP) were subcutaneously injected into C57BL/6 J mice to induce unilateral xenograft hindleg tumors ($1 \times 10^6$ cells per unilateral inoculation). PDX and MOC1 tumor volumes were monitored twice a week starting on day 7 after tumor inoculation, and mice underwent experimental procedures when masses reached a size between 150 and 250 mm$^3$. All cell lines used in vivo tested negative for mycoplasma contamination, as required by Emory University School of Medicine's IACUC. Specifically, PCR-based mycoplasma testing was performed on FaDu, MOC1, and PDX1 cell lines, since these 3 lines were used in animal studies. Authentication of cell lines was performed by the commercial provider ATCC in the case of FaDu and MOC1 cells, per its commercial guidelines, and the National Cancer Institute (NCI) in the case of the PDX cells, per the PDMR guidelines.

## Cell isolation and staining

Cells were isolated from digested tissues, which were harvested 16 hours after intratumoral injection with LNPs, unless otherwise noted. Mice were perfused with 20 mL of 1X PBS through the right atrium when harvesting non-tumor tissues. Tumors or livers were finely minced and transferred into a digestive enzyme solution with collagenase type I (Sigma-Aldrich), collagenase IV (Sigma-Aldrich), collagenase XI (Sigma-Aldrich), and hyaluronidase (Sigma-Aldrich), and incubated at 37 °C with controlled shaking at 550 rpm for 45 minutes. Cell suspensions were then filtered through a 70 µm mesh. Next, cells were stained to identify specific cell populations and gate out lysed cells and red blood cells. The target CD47+ aVHH+ cell populations were then sorted using the BD FACS Fusion cell sorters in the Georgia Institute of Technology Cellular Analysis Core. The antibody clones used were anti-CD31 (390, BioLegend), anti-CD45.2 (104,

BioLegend), anti-hCD47 (CC2C6, BioLegend), and MonoRab™ rabbit anti-camelid VHH antibody iFluor647 (A01994, GenScript), all at 1:500 dilution ratios in 10% fetal bovine serum (FBS) in PBS solutions, where the single-cell suspension samples were dissolved for flow cytometry. Representative flow gates are located in Supplementary Fig. 1, Supplementary Fig. 13.

## PCR amplification

We amplified and prepared samples for sequencing following a one-step PCR protocol[70]. Specifically, 1 µL of primers (5 mM for final reverse/forward, 0.5 mM for base forward) were added to 5 µL of Kapa HiFi 2X master mix and 4 µL template DNA/water. If no clear bands were produced by the PCR reaction, we adjusted and optimized the primer concentrations, DNA template input, PCR temperature, and number of cycles for individual samples.

## Deep sequencing and normalization

Illumina deep sequencing was performed on an Illumina MiniSeq™. Primers were designed based on Nextera XT adapter sequences. Counts for each particle, per tissue, were normalized to the barcoded LNP mixture injected into each mouse. This 'input' DNA provided DNA counts and was used to normalize DNA counts from cells and tissues. Sequencing results were processed using a custom Python-based tool to extract raw barcode counts from individual samples. Raw counts were then normalized with an R script prior to further assessment. Statistical analysis was conducted using GraphPad Prism 8.

## Whole organ imaging

Tissues were isolated 24 hours after administration of LNPs, unless otherwise noted. To measure luminescence, mice were sacrificed and organs collected, followed by treatment with Nano-Glo Luciferase Assay Substrate (Promega) for 5 minutes before being placed on solid black paper for imaging. Luminescence was measured using an IVIS system (PerkinElmer) and quantified using Living Image software (PerkinElmer).

## Single-cell RNA sequencing (scRNA-seq)

Sixteen hours following IT injection of aVHH mRNA-encapsulating LNP[28] or LNP$^{IT}$ to PDX tumors, masses were dissected from the animals. After tissue digestion, cells were resuspended in RoboSep buffer (Stemcell Technologies) for further processing. Whole transcriptome analyses were performed using the BD Rhapsody Single-Cell Analysis System (BD Biosciences), following the manufacturer's protocol. Briefly, dead cells and red blood cells (RBCs) were depleted by using EasySep™ dead cell (Annexin V) and RBC (anti-TER119) removal kit (Stemcell Technologies). Cell viability and numbers were recorded for each sample, followed by tagging with TotalSeq™ anti-human Hashtag antibody (5 µg/mL; [TotalSeq-A0251 (BioLegend 394601), TotalSeq-A0252 (BioLegend 394603), TotalSeq-A0253 (BioLegend 394605), TotalSeq-A0257 (BioLegend 394613), TotalSeq-A0258 (BioLegend 394615), TotalSeq-A0259 (BioLegend 394617)]) and oligo-tagged anti-VHH antibody (5 µg/mL; A01860, GenScript). For the preparation of oligo-tagged anti-VHH antibody, 5′ DBCO-modified oligonucleotide (CCTTGGCACCCGAGAATTCCAAA GTATGCCCTACGABAAAAAAAAAAAAAAAAAAAAAAAAAAA*A*A, where * indicates phosphothioate bonds; GenScript) was conjugated to azide-modified rabbit anti-camelid VHH antibody (clone 96A3F5; GenScript) by click chemistry (GenScript). Samples were then pooled at the same ratio and a BD Rhapsody cartridge loaded with 40,000 cells. cDNA libraries were prepared using the BD Rhapsody Whole Transcriptome Analysis Amplification Kit following the BD Rhapsody System mRNA Whole Transcriptome Analysis (WTA) and Sample Tag Library Preparation protocol (BD Biosciences). Final libraries were quantified using a Qubit Fluorometer and the quality checked with BioAnalyzer (Agilent) by size distribution. The data were processed

using STARsolo (v 2.9.7) for the RNA mapping and counting[71]. All samples were mapped to GRCh38.p14, and only exonic regions were counted. Output files were next loaded into Seurat (v 4.0.4) and cells log normalized to a scale factor of 10,000, then scaled using a linear transformation[72]. DoubletFinder (v3) (Ref. [73]), which is an algorithm for scRNA-seq datasets that predicts doublets according to each real cell's proximity in gene expression space to artificial doublets resulting from averaging the transcriptional profile of randomly chosen cell pairs, was used to identify doublets. This was followed by PCA dimensional reduction and t-SNE clustering and further analyzed in BBrowserX[74]. In BBrowserX, gene expression profiles were compared within cell types of interest. Reactome pathway analysis was performed using Reactome database 85[52]. The pathway expression levels are shown as z-score normalized values.

## PNP enzymatic quantification assay

PDX tumors harvested from the flanks of *NSG* mice were flash frozen and stored at −80 °C until preparation of extract. Crude extracts were prepared[53] By homogenizing fixed tumor samples in 0.01 M HEPES buffer and then incubated on ice for 15 min. Samples were then sonicated three times for 30 seconds each, centrifuged at 100,000 $g$ for 60 minutes, and dialyzed with 50 volumes of 0.1 M HEPES buffer (1 mM DTT and 20% glycerol at pH 7.4). These crude extracts were then incubated at 25 °C with 50 mM potassium phosphate, 100 μM 6-methylpurine-2′-deoxyriboside (MeP-dR), 100 mM HEPES buffer (pH 7.4), and an amount of enzyme that yielded a linear signal during the incubation period. Reactions were stopped by boiling. The formation of 6-methylpurine (MeP) was monitored using reverse-phase HPLC (PerkinElmer). The level of MeP produced from MeP-dR is directly related to the amount of PNP expressed in the PDX cells. The PNP activity is presented as nmoles of MeP produced per mg protein per hour.

## Spin concentration of PNP mRNA-carrying LNP[IT]

LNP[IT]-PNP formulated in the NanoAssemblr Ignite was diluted in molecular grade sterile 10 mM tris buffer pH 7.4 (Calbiochem) at a 1:40 ratio of LNP:tris buffer dilution. This was then poured into Amicon Ultra-15mL centrifugal filter tubes of 100 kDa membranes (Millipore) and centrifuged at 1,000 G for 30-minute cycles until the whole LNP in tris solution was brought to the desired concentration.

## In vitro cell killing assay

MOC1 ($3 \times 10^4$ cells/well), FADU ($2 \times 10^5$ cells/well) or PDX ($8 \times 10^4$ cells/well) cells were seeded into 24-well plates, and LNP[IT] carrying PNP mRNA was added 24 h after plating at 1 μg/well. MeP (6-methylpurine) or MeP-dR (9-(2-deoxy-beta-D-ribofuranosyl)-6-methylpurine) was added at 100 μM within 20 h after LNP transfection. Cells were monitored for 4 days and stained with 0.1% crystal violet to evaluate cell survival. During the assay, dead or detached cells were washed away, while attached live cells were stained with crystal violet. Clear wells indicate >95% cell killing.

## In vitro PNP activity measurement

MOC1 (Kerafast, catalog #EWL001-FP), FADU (ATCC, catalog #HTB-43), or PDX cells (NCI PDMR, PDX1 – 328373-195-R-J1-PDC, PDX2 – 929823-356-R-J2-PDC, PDX3 – 958767-090-R-J1-PDC, PDX4 – 845751-090-R-J2-PDC) were seeded into 6-well plates, and LNP[IT] carrying PNP mRNA at 4.5 μg/well was added 24 h after plating. The following day, cells were incubated for 4 hours with 100 μM MeP-dR (substrate for *E. coli* PNP, see above), and formation of MeP (cytotoxin) was measured using reverse phase HPLC (PerkinElmer).

## Tumor volume measurement and clinical scoring

Tumor volume measurements were performed using a digital caliper (Fisherbrand) and initiated 7 days after PDX tumor inoculation.

Width (W, in mm) and length (L, in mm) measurements were taken, and tumor volume (V, in mm³) was calculated using the following formula:

$$V = \frac{L \, x \, W^2}{2}$$

Once tumor sizes reached between 150 and 250 mm³ in size, animals were divided into experimental groups and treated as noted in the text. Tumor volume and body weight were measured for each mouse twice a week for the remainder of the study or until the animals reached endpoint conditions and were euthanized. Clinical scores were calculated based on the tumor burden (size and ulceration) and body condition and mobility (Supplementary Fig. 9). Tumor diameter was measured, and a tumor burden score (TBS) was assigned following Emory University's IACUC policy 304. A TBS of 0 was assigned when the tumor diameter was less than 18 mm, TBS 1 when the tumor diameter was between 18 and 20 mm, and TBS 3 when the diameter was larger than 20 mm. When the maximal TBS of 3 was reached, the animal was euthanized.

## Statistics & reproducibility

All experiments were done at least in triplicate ($n = 3$). For all tumor regression studies, $n = 6$ was used for each group and mice were randomized, ensuring that the average tumor size was similar across all groups when beginning treatment. PDX in vivo studies were performed twice to ensure the reproducibility of the therapeutic approach, which was validated further in other in vivo tumor models. No statistical method was used to predetermine sample size. No data were excluded from the analyses. The Investigators were not blinded to allocation during experiments and outcome assessment. All measurements were taken from distinct samples; the same sample was NOT measured repeatedly in any instance.

## Reporting summary

Further information on research design is available in the Nature Portfolio Reporting Summary linked to this article.

## Data availability

The scRNA-seq data generated in this study have been deposited in the SRA database [https://www.ncbi.nlm.nih.gov/sra] under BioProject ID PRJNA1090507, BioSamples accession numbers SAMN40568326 and SAMN40568327, SRA numbers SRS20810744 [https://www.ncbi.nlm.nih.gov/biosample/40568326] and SRS20810746 [https://www.ncbi.nlm.nih.gov/biosample/40568327]. Other data generated in this study are provided in the main figures, Supplementary Information or the Source data file. Source data are provided with this paper.

## Code availability

All code used to analyze the data is available (https://github.com/Jack-Feldman/barcode_count).

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

## Acknowledgements

The graphical schematics were generated with BioRender. We would like to thank the NCI Patient-Derived Models Repository (PDMR; https://pdmr.cancer.gov; NCI-Frederick, Frederick National Laboratory for Cancer Research, Frederick, MD) for help obtaining materials. This research was funded by the National Institutes of Health (R01DE026941, awarded to E.J.S. and J.E.D.) and the Georgia Research Alliance.

## Author contributions

S.G.H., P.J.S., E.J.S. and J.E.D. conceptualized and designed experiments, which were performed by S.G.H., L.L., R.R., Y.H., A.R., H.K., R.Z., B.R.A., M.G.R., M.P.L., D.L., H.E.P., E.S.E., A.J.D.S.S., A.S. and A.L.; data was analyzed by S.G.H., R.R., Y.H., M.G.R., H.K., E.J.S. and J.E.D.; S.G.H., K.E.T., E.J.S. and J.E.D. wrote the paper, which was reviewed by all other authors.

## Competing interests

S.G.H., R.R., B.R.A., M.P.L., P.J.S., E.S. and J.E.D. have filed intellectual property related to this manuscript (PCT/US2023/033245). J.E.D. is an advisor to Readout Capital, Edge Animal Health, and Nava Therapeutics. E.J.S. has an ownership interest in PNP Therapeutics, Inc., and serves on the Board of Directors for the company, which develops products used in cancer research. Dr. Sorscher is also an inventor of technology being evaluated in studies described by this report. The terms of this arrangement for Dr. Sorscher have been reviewed and approved by Emory University in accordance with its conflict-of-interest policies. All other authors declare no conflict of interest.
