## [Transparent Peer Review file · Nature Communications]

Nanoparticle delivery of a prodrug-activating bacterial enzyme leads to anti-tumor responses

Corresponding Author: Professor James Dahlman

Version 0:

Reviewer comments:

Reviewer #1

(Remarks to the Author)

This study presents a compelling investigation involving the development of a Lipid Nanoparticle (LNP) encapsulated bacterial enzyme, PNP, designed to activate a purine analogue, fludarabine as a therapeutic strategy for head and neck squamous cell carcinoma (HNSCC). The authors systematically screened a panel of LNPs to identify the two most effective LNPs in delivering this enzyme, as demonstrated by a FaDu-xenograft model. Minimal off-target effects were observed in this model. Subsequently, the authors utilized a Patient-Derived Xenograft (PDX) model by inoculating HNSCC tumors into NSG mice, revealing that a specific LNP, LNP-IT, exhibited optimal mRNA delivery to tumor cells. The study further employed single-cell RNA sequencing (scRNA-seq) to investigate the impact of tumor heterogeneity on LNP delivery, revealing varied mRNA delivery in different tumor clusters. Finally, the study validated the effectiveness of LNP-delivered PNP followed by fludarabine in treating HNSCC in the PDX model. Overall, this study underscores the potential of intratumoral delivery of therapeutic mRNA in advanced HNSCC treatment.

The study may benefit from the following suggested improvements to enhance the thoroughness. To minimize the usage of experimental mice, the following experiments may only validate the effect of LNP-IT.

1. The authors acknowledge that tumor heterogeneity may affect LNP-mediated mRNA delivery (Fig. 4, scRNA-seq experiment). However, the study primarily employed one HNSCC cell line-derived xenograft and one PDX model for testing LNP-mediated mRNA delivery and validated therapeutic impact solely in the PDX model. Considering the heterogeneity within HNSCC, utilizing different HNSCC xenograft models derived from various HNSCC cell lines representing distinct HNSCC phenotypes (e.g., pro-metastatic tumor such as SAS cells vs. pro-growing tumor such as FaDu cells) would enhance the robustness of the findings.

2. Regarding the PDX model, the study employed a single PDX. Expanding the study to include multiple PDX models from different patients would enrich the investigation. Additionally, providing details about the xenografted tumor characteristics, including patient age, gender, HPV status, and tumor stage, would be informative. Particularly, recruiting HPV-positive vs. HPV-negative tumors for deriving PDX models is recommended, given the distinct characteristics displayed by these tumor types.

3. To better comprehend the role of immune cells in LNP-mediated RNA delivery, the authors may consider investigating a syngeneic HNSCC model. Since the current models involve immunodeficient mice, extending the study to incorporate an immunocompetent syngeneic HNSCC model would provide valuable insights into the interaction between LNPs and the immune system.

4. The analysis of transcriptional responses to LNPs carrying mRNA is intriguing. To validate these findings, conducting in vitro experiments with cell lines would be an excellent extension, further confirming the robustness and applicability of the observed transcriptional responses.

Reviewer #2

(Remarks to the Author)

This manuscript describes an RNA-derived enzyme pro-drug therapy with demonstrated efficacy in pre-clinical models of head and neck squamous cell carcinoma (HNSCC). The key novelty is that the mRNA delivered encodes for a bacterial enzyme, purine nucleoside phosphorylase (PNP), which can cleave the nucleoside fludarabine to the cytotoxic agent 2-

fluoroadenine (F-Ade). After intratumoural injection of the mRNA via a lipid-derived delivery system, subsequent administration of fludarabine is expected to lead to in situ formation of F-Ade wherever the bacterial PNP is expressed. Since human PNP does not act on fludarabine, only where the E coli PNP is present will F-Ade be formed, and this is controlled by the intratumoural location of the injected mRNA. Further control is potentially possible because fludarabine is administered as its soluble 5-O-phosphorylated form, and is converted in the body where appropriate phosphorylases are present.

The data in the paper convincingly show that multiple LNP formulations deliver mRNA effectively to a range of cancer cells in vivo. Initial reporter assays employed NanoLuc and a glycosylphosphatidylinositol (GPI) anchored camelid single-variable domain on a heavy chain antibody (aVHH), in xenografted HNSCC tumours in mice. Luciferase expression was used as a screen for the formulations and introduction of DNA barcodes enabled identification of which formulations most effectively induced aVHH expression. A key finding was that two lipids, derived from the cationic C12-200 backbone, were most effective. These were KB-S13, formed by reacting C12-200 with S-1,2-epoxypentadecane and KB-R12 with R-1,2-epoxytetradecane. Further assays involved intratumoural injection of formulations from these 2 lipids with mRNA for NanoLuc and the species-agnostic reporter aVHH as before, but this time monitoring for off-target effects as well as expression in tissue. Experiments in PDX cancer models demonstrated that the KB-R12 formulation was more effective in transfecting cells in the tumours than the KB-S13 formulation, and single-cell RNA-seq was used to determine formulation efficacy in gene-clustered cells from the heterogenous tumour populations. Following the biomarker assays the LNPIT formulation was prepared with an mRNA encoding for PNP, to test the hypothesis that expression of PNP in situ with a subsequent dosing of fludarabine phosphate could have anti-tumour effects. PNP expression was observed, as expected, in the PDX tumours, and administration of fludarabine phosphate as a single dose after LNPIT injection resulted in tumour volume reduction and overall survival benefit in the mice.

The manuscript represents an advance over current methodologies and is both novel and significant. There are caveats to the study, as the authors note, but the data are robust and the conclusions drawn from them are reasonable. Publication is recommended subject to the revisions below.

1. Preparation and formulation of LNPs is highly dependent on flow rates, mixing speeds, and volumes and concentrations of components. The experimental section (page 12) refers to a paper from the Cullis group in 2013, who used a staggered herringbone mixer (SHM), but it isn't clear exactly how the mixing was done in this case. Flow rates are given for the scale-up but not the main formulation studies. In addition, in the ESI, Table 1 of the different formulations, the caption does not indicate whether the compositions are given as v/v, molar %, and nor is the 'mass ratio' defined. The ESI should be updated to include volumes and concentrations of reagents, clear definitions of the parameters given in the table, and clarification of the mixing conditions and flow rates used in the fluidics system.
2. For the 2 formulation LNPIT and LNP28 there is no significant difference between protein expression from mRNA NanoLuc and aVHH by these formulations in FaDu-derived tumors. However, for the PDX models, LNPIT was much more effective, which the authors note, but for which they do not suggest a reason. Could this be due to the difference in ECM or organisation in patient-derived tumours as opposed to the FaDu xenografts?
3. In the authors' recent study (Journal of Controlled Release 2023, 353, 270–277) the most effective C12-200 lipid for a Cre-encoding mRNA in Ai14 mice (engineered with a CAG-Lox-Stop-Lox tdTomato construct) was one in which an S-epoxide was used. In the prior case a ~ 3-fold higher expression for the S-LNP was obtained over the R-isomer. The authors attributed this to better tolerability of the S-lipid in the mice rather than due to other structural parameters such as particle size, charge, stability. For this manuscript, the R-lipid was much more effective – why is this?

Reviewer #3

(Remarks to the Author)

The mRNA vaccine has achieved great success and has also been attracted to be used in the treatment of various diseases, including the anti-tumor field. In this work, the authors developed the combinatorial application of mRNA and prodrugs to achieve antitumor responses. Fludarabine is a prodrug that does not have antitumor effects but can effectively produce a cytotoxic form after catalysis by purine nucleoside phosphorylase (PNP). The authors screened a series of LNPs that could specifically deliver PNP mRNA in tumor tissue but not other tissues, thus reducing the off-target toxicity of fludarabine. Overall, this is one of the attractive applications of mRNA technology, and most results support the conclusions well. Some comments are listed below.

1. In Figure 1c, C12-200 LNP showed the most efficient effect on Luc mRNA delivery after i.t. injection. It is recommended to use this original C12-200 LNP as a positive control to compare with LNP28 and LNPIT for i.t. delivery of mRNA.
2. PNP is a foreign protein produced from mRNA. Will the injection trigger any unexpected immune responses?
3. 2-Fluoroadenine is the ultimate antitumor effector. It is recommended to detect 2-fluoroadenine abundance in tumors when treated by PNP mRNA and fludarabine combination.
4. According to point 3, it is recommended to add 2-fluoroadenine group as a positive control for anti-tumor response.

Reviewer #4

(Remarks to the Author)

Summary: In this manuscript, the authors demonstrate that lipid nanoparticles (LNPs) are an efficient mRNA delivery system for the treatment of solid tumors. To design an optimized LNP for intratumoral delivery in vivo, they first used a pooled LNP library with a DNA barcoding system for multiplexing and identified a lead LNP, named LNP^{IT}, with enhanced transfection efficiency and minimal off-targets in both HNSCC xenograft and PDX mouse models. Using scRNA-seq analysis, they

showed that cancer cells transfected with LNP^{IT} exhibited altered RNA and protein processing. Finally, by combining their LNP delivery system with mRNA encoding purine nucleoside phosphorylase, which converts fludarabine to 2-fluoroadenine (cytotoxic drug), they showed that LNP^{IT}-mRNA treatment leads to anti-tumor responses.

The present manuscript is well written and nicely demonstrates that the LNP-mRNA delivery system is an effective approach for the treatment of solid tumors. However, there are several important points that should be addressed to substantiate the authors' claims.

Major points:

1. Figure 4: The corresponding scRNA-seq analysis seems superficial and requires a more in-depth analysis to convincingly support the main claims. First, it is not clear how the authors quantified aVHH protein expression (Line 208) in Figure 4c and d. Did they use CITE-seq? If Figure 4c and d show aVHH mRNA expression, then Line 208 should be corrected. Second, the condition information of each cell (LNP^{IT}-mRNA or PBS control) should be displayed. As a negative control, the absence of aVHH mRNA expression in the PBS control should be shown. Third, the authors should analyze cellular and molecular alterations induced by LNP^{IT}-mRNA transfection in all cell types including stromal and immune cells. Fourth, how do the authors distinguish aVHH⁺ from aVHH⁻ cells? If there are aVHH⁺ cells in the PBS control due to ambient RNAs, the same DEG and pathway analysis should be performed in the PBS condition as a negative control. A more detailed method for scRNA-seq data analysis including DEG and pathway analysis should be provided. Fifth, the authors should characterize the molecular features of the cancer cell clusters in more detail. Finally, some up- and down-regulated genes in Figure 4e should be experimentally validated.

Minor points:

1. Define "LNP" and "FaDu" in the abstract.

Version 1:

Reviewer comments:

Reviewer #1

(Remarks to the Author)

The authors satisfactorily addressed the reviewers' queries. Regarding the cell lines, PDX and animal models used in this study, they incorporated new findings from multiple HNSCC cell lines and supplemented the study with additional data from PDX and syngeneic HNSCC models. Additionally, they successfully validated the genes responsive to LNPs in a second PDX line in vitro. Overall, the manuscript has significantly improved, and I recommend it for publication.

Reviewer #2

(Remarks to the Author)

I am happy to recommend publication of this manuscript. The authors have responded well to the comments which I made, and to those of other reviewers in areas of expertise that I feel able to evaluate with sufficient knowledge.

Reviewer #3

(Remarks to the Author)

The author has addressed my concerns. I recommend accept this manuscript.

Reviewer #4

(Remarks to the Author)

The authors have satisfactorily addressed my comments.

Response to reviewer comments for Sebastian G Huayamares et al.

We want to thank the editorial team and reviewers for taking the time to evaluate the manuscript. Our responses are in blue and new writing / data is in green.

Reviewer 1 This study presents a compelling investigation involving the development of a Lipid Nanoparticle (LNP) encapsulated bacterial enzyme, PNP, designed to activate a purine analogue, fludarabine as a therapeutic strategy for head and neck squamous cell carcinoma (HNSCC). The authors systematically screened a panel of LNPs to identify the two most effective LNPs in delivering this enzyme, as demonstrated by a FaDu-xenograft model. Minimal off-target effects were observed in this model. Subsequently, the authors utilized a Patient-Derived Xenograft (PDX) model by inoculating HNSCC tumors into NSG mice, revealing that a specific LNP, LNP-IT, exhibited optimal mRNA delivery to tumor cells. The study further employed single-cell RNA sequencing (scRNA-seq) to investigate the impact of tumor heterogeneity on LNP delivery, revealing varied mRNA delivery in different tumor clusters. Finally, the study validated the effectiveness of LNP-delivered PNP followed by fludarabine in treating HNSCC in the PDX model. Overall, this study underscores the potential of intratumoral delivery of therapeutic mRNA in advanced HNSCC treatment.

The study may benefit from the following suggested improvements to enhance the thoroughness. To minimize the usage of experimental mice, the following experiments may only validate the effect of LNP-IT.

1. The authors acknowledge that tumor heterogeneity may affect LNP-mediated mRNA delivery (Fig. 4, scRNA-seq experiment). However, the study primarily employed one HNSCC cell line-derived xenograft and one PDX model for testing LNP-mediated mRNA delivery and validated therapeutic impact solely in the PDX model. Considering the heterogeneity within HNSCC, utilizing different HNSCC xenograft models derived from various HNSCC cell lines representing distinct HNSCC phenotypes (e.g., pro-metastatic tumor such as SAS cells vs. pro-growing tumor such as FaDu cells) would enhance the robustness of the findings. We agree re: the need to see the effect in multiple models. We therefore measured the anti-tumor response in two *in vivo* models: (i) FaDu, a human HNSCC xenograft in Nu/J mice, and (ii) MOC1, a murine syngeneic HNSCC line in immunocompetent mice. We also measured cell killing in three new patient-derived xenograft (PDX) lines *in vitro* (termed PDX2, 3, and 4). We added these data into Figure 6:

Since HNSCC tumors are heterogenous, we tested therapeutic effect in two additional *in vivo* models: FaDu tumors in immunocompromised NU/J mice (**Fig. 6a**) and MOC1 murine oral cancers (syngeneic HNSCC) in immunocompetent C57BL/6 mice (**Fig. 6b**). Tumors treated with the combination therapy were significantly smaller than those in the control groups. Given that MOC1 tumors are characterized by increased MHC-I expression and CD8⁺ T cell infiltration into the tumor microenvironment⁵² while FaDu tumors were grown in NU/J mice with T cell deficiency⁵³, these data provide additional evidence that the approach is active in a number of different host immunologic contexts.

To evaluate therapeutic response in distinct human cancer cells, we also performed an *in vitro* cell killing assay (**Fig. 6c**). We plated six different HNSCC cell lines: human FaDu, murine MOC1, and four HNSCC patient-derived xenograft lines, termed PDX1, PDX2, PDX3, and PDX4 (**Supplementary Fig. 12a**). We treated cells with LNP^{IT}-PNP and MeP-dR (a nucleoside analogue of fludarabine that has served as a prototype for *in vitro* testing). We showed the expected PNP and nucleoside dependent cell killing and also observed time-

dependent conversion from prodrug to oncolytic drug (**Supplementary Fig. 12b**). The negative controls, LNP-PNP^{IT} alone, MeP-dR alone, and no treatment behaved as expected. In contrast, cells treated with the positive control MeP (the toxic base released following MeP-dR hydrolysis by PNP) led to the expected ablation of cells in culture.

Figure 6 | LNP^{IT}-PNP and fludarabine phosphate combination therapy has anti-tumor effects validated across various HNSCC preclinical models. Tumor growth studies in two additional *in vivo* HNSCC models: **a**, FaDu, human HNSCC xenografts inoculated in immunocompromised NU/J mice, and **b**, MOC1, syngeneic HNSCC murine tumors inoculated in immunocompetent C57BL/6 mice. Tumor volume was compared to the control groups (N=6 per cohort) using a two-way ANOVA Tukey's multiple comparisons test (mean +/- SD; * p<0.05, *** p<0.001, **** p<0.0001). **c**, An *in vitro* assay to assess the anti-tumor cell effects of combination therapy in the syngeneic MOC1 murine line, FaDu human tumor cells, and four HNSCC patient-derived xenograft lines (PDX1, PDX2, PDX3, and PDX4). MeP-dR is an analogue of fludarabine phosphate (prodrug), used as a prototype compound for showing PNP activity. MeP is the toxic cleavage product of MeP-dR following PNP treatment, and serves as a positive control.

We also described the *in vitro* cell killing assay in the Methods:

***In vitro* cell killing assay.** MOC1 (3x10⁴ cells/well), FADU (2x10⁵ cells/well) and PDX (8x10⁴ cells/well) cells were seeded into 24-well plates, and LNP^{IT} carrying

PNP mRNA was added 24 h after plating at 1 $\mu\text{g}/\text{well}$. MeP (6-methylpurine) or MeP-dR (9-(2-deoxy-beta-D-ribofuranosyl)-6-methylpurine) was added at 100 μM within 20 h after LNP transfection. Cells were monitored for 4 days and stained with 0.1% crystal violet to evaluate cell survival. During the assay, dead or detached cells were washed away, while attached live cells were stained with crystal violet. Clear wells indicate >95% cell killing.

2. Regarding the PDX model, the study employed a single PDX. Expanding the study to include multiple PDX models from different patients would enrich the investigation. Additionally, providing details about the xenografted tumor characteristics, including patient age, gender, HPV status, and tumor stage, would be informative. Particularly, recruiting HPV-positive vs. HPV-negative tumors for deriving PDX models is recommended, given the distinct characteristics displayed by these tumor types. [We added the NCI PDMR PDX information \(https://pdmr.cancer.gov/database/default.htm\)](https://pdmr.cancer.gov/database/default.htm) to **Supplementary Fig. 12a**.

a

	PDX1	PDX2	PDX3	PDX4
Patient ID	328373	929823	958767	845751
Disease Body Location	Head and Neck	Head and Neck	Head and Neck	Head and Neck
OncoTree Code	HNSC	OSCS	OSCS	OSCS
Tissue Type	Resection	Resection	Resection	Resection
Tissue Collected	Neck (L)	Tongue	Tongue (Left, Lateral)	Neck (Bilateral)
Provided Tissue Origin	Primary	Primary	Primary	Primary
Date of Diagnosis	06/2014	12/2014	02/2015	01/2005
Collection Date	07/2014	12/2014	03/2015	03/2015
Age of Sampling	88	43	59	73
Gender	Male	Male	Male	Female
Human Pathogen Testing Summary	Negative	Negative	Negative	Negative
Grade/Stage Information Available	None Provided	None Provided	TMN (Pathological)	None Provided
Has Known Metastatic Disease	Not Reported	Not Reported	Yes. Tumor Grade/ Stage: pT4aN2b	Not Reported

Supplementary Fig. 12 | PDX models and prodrug kinetics. a, PDX models used in this work. **b**, Conversion kinetics of fludarabine into the activated oncolytic form by LNP^{IT}-PNP in all HNSCC models studied.

3. To better comprehend the role of immune cells in LNP-mediated RNA delivery, the authors may consider investigating a syngeneic HNSCC model. Since the current models involve immunodeficient mice, extending the study to incorporate an immunocompetent syngeneic HNSCC model would provide valuable insights into the interaction between LNPs and the immune system. We observed anti-tumor activity in the syngeneic MOC1 model in wildtype C57BL/6 mice (please see response to comment 1 above).

4. The analysis of transcriptional responses to LNPs carrying mRNA is intriguing. To validate these findings, conducting *in vitro* experiments with cell lines would be an excellent extension, further confirming the robustness and applicability of the observed transcriptional responses. We validated several genes identified in our *in vivo* scRNA-seq experiment in a second PDX line transfected with LNP^{IT} *in vitro* and added the new data (Supplementary Fig. 8c).

Of these, 14 were directly associated with mRNA translation into protein (R-HSA-72766) and stemmed directly from the core of the metabolic pathway tree (Biological Process GO:0019538), consistent with previously reported transcriptional responses to mRNA-carrying LNPs³⁶ (**Fig. 4f,g, Supplementary Fig. 8b**). Some of the most upregulated genes found in this PDX model *in vivo* were also validated in a different PDX model (PDX2) transfected with LNP^{IT}

(Supplementary Fig. 8c). These data provide an early line of evidence that HNSCC cancer cells respond to LNPs carrying mRNA in part by altering genes related to the manufacture and processing of RNA and protein.

b

Reactome pathway related to mRNA translation into protein

1. Translation
2. SRP Dependent co-translational protein targeting to membrane
3. Eukaryotic translation termination
4. Eukaryotic translation initiation
5. L13a-mediated translational silencing of ceruloplasmin expression
6. Cap-dependent translation initiation
7. Ribosomal scanning and start codon recognition
8. GTP hydrolysis and joining of 60S ribosomal subunit
9. Formation of ternary complex, and subsequently, the 43S complex
10. Formation of a pool of free 40S subunit
11. Activation of the mRNA upon binding of the cap-binding complex and eIFs and subsequent binding to 43S
12. Translation initiation complex formation
13. Eukaryotic translation elongation
14. Peptide chain elongation

Reactome pathway related to other protein metabolic processes

- | | |
|---|---|
| 15. Amyloid fiber formation | 22. Ovarian tumor domain proteases |
| 16. Post-translational protein modification | 23. Metalloprotease DUBs |
| 17. Protein ubiquitination | 24. Josephin domain DUBs |
| 18. E3 ubiquitin ligases ubiquitinate target proteins | 25. Asparagine N-linked glycosylation |
| 19. Synthesis of active ubiquitin: roles of E1 and E2 enzymes | 26. N-glycan trimming in the ER and Calnexin/Calreticulin cycle |
| 20. Deubiquitination | 27. Calnexin/Calreticulin cycle |
| 21. UCH proteinases | 28. ER Quality Control Compartment (ERQC) |

c

PDX2 transfected with LNP^{IT}

Supplementary Fig. 8 | Reactome pathway analysis for scRNA-seq data from LNP^{IT}. [...] **c,** Some of the top up-regulated genes identified *in vivo* in the PDX tumors transfected by LNP^{IT} were validated *in vitro* in a second PDX line, PDX2, via qRT-PCR.

Reviewer 2 This manuscript describes an RNA-derived enzyme pro-drug therapy with demonstrated efficacy in pre-clinical models of head and neck squamous cell carcinoma (HNSCC). The key novelty is that the mRNA delivered encodes for a bacterial enzyme, purine nucleoside phosphorylase (PNP), which can cleave the nucleoside fludarabine to the cytotoxic agent 2-fluoroadenine (F-Ade). After intratumoural injection of the mRNA via a lipid-derived delivery system, subsequent administration of fludarabine is expected to lead to in situ formation of F-Ade wherever the bacterial PNP is expressed. Since human PNP does not act on fludarabine, only where the E coli PNP is present will F-Ade be formed, and this is controlled by the intratumoural location of the injected mRNA. Further control is potentially possible because fludarabine is administered as its soluble 5-O-phosphorylated form, and is converted in the body where appropriate phosphorylases are present.

The data in the paper convincingly show that multiple LNP formulations deliver mRNA effectively to a range of cancer cells *in vivo*. Initial reporter assays employed NanoLuc and a glycosylphosphatidylinositol (GPI) anchored camelid single-variable domain on a heavy chain antibody (aVHH), in xenografted HNSCC tumours in mice. Luciferase expression was used as a screen for the formulations and introduction of DNA barcodes enabled identification of which formulations most effectively induced aVHH expression. A key finding was that two lipids, derived from the cationic C12-200 backbone, were most effective. These were KB-S13, formed by reacting C12-200 with S-1,2-epoxypentadecane and KB-R12 with R-1,2-epoxytetradecane. Further assays involved intratumoural injection of formulations from these 2 lipids with mRNA for NanoLuc and the species-agnostic reporter aVHH as before, but this time monitoring for off-target effects as well as expression in tissue. Experiments in PDX cancer models demonstrated that the KB-R12 formulation was more effective in transfecting cells in the tumours than the KB-S13 formulation, and single-cell RNA-seq was used to determine formulation efficacy in gene-clustered cells from the heterogeneous tumour populations. Following the biomarker assays the LNPIT formulation was prepared with an mRNA encoding for PNP, to test the hypothesis that expression of PNP *in situ* with a subsequent dosing of fludarabine phosphate could have anti-tumour effects. PNP expression was observed, as expected, in the PDX tumours, and administration of fludarabine phosphate as a single dose after LNPIT injection resulted in tumour volume reduction and overall survival benefit in the mice.

The manuscript represents an advance over current methodologies and is both novel and significant. There are caveats to the study, as the authors note, but the data are robust and the conclusions drawn from them are reasonable. Publication is recommended subject to the revisions below.

1. Preparation and formulation of LNPs is highly dependent on flow rates, mixing speeds, and volumes and concentrations of components. The experimental section (page 12) refers to a paper from the Cullis group in 2013, who used a staggered herringbone mixer (SHM), but it isn't clear exactly how the mixing was done in this case. Flow rates are given for the scale-up but not the main formulation studies. In addition, in the ESI, Table 1 of the different formulations, the caption does not indicate whether the compositions are given as v/v, molar %, and nor is the 'mass ratio' defined. The ESI should be updated to include volumes and concentrations of reagents, clear definitions of the parameters given in the table, and clarification of the mixing conditions and flow rates used in the fluidics system. We added the flow rate and formulation details.

Nanoparticle Formulation. Nanoparticles for *in vivo* screening were formulated using a microfluidic device as previously described³⁹ at a flow rate ratio of 3:1 of nucleic acid:lipid phases. Larger batches of LNP^{IT} with PNP mRNA for the preclinical tumor regression studies were formulated using the NanoAssemblr Ignite (Precision Nanosystems). DNA barcodes and/or mRNA were diluted in 10 mM citrate buffer (Teknova). DNA barcodes were purchased from IDT. PEGs, cholesterol, and helper lipids were diluted in 100% ethanol and purchased from Avanti Lipids. Citrate and ethanol phases were combined in a microfluidic device or the NanoAssemblr Ignite using glass (Hamilton Company) or plastic (BD) syringes, respectively, at a flow rate ratio of 3:1.

Supplementary Fig. 1 | LNP screening compositions. **a**, Composition (in molar %), diameter (nm), and PDI of the initial bioluminescence-based screen of LNPs with Dlin-MC3-DMA, cKK-E12, SM-102, and C12-200 as ionizable lipids, intratumorally injected to FaDu HNSCC tumors in NU/J mice. **b**, Composition (in molar %) and mass ratios (w/w of lipid/nucleic acid) of the 64 LNPs formulated, out of which 44 had diameters between 50 and 200 nm and were injected to FaDu HNSCC tumors in NU/J mice. The winner LNPs (28 and 49/LNP^{IT}) are

highlighted in a red box. Reagent concentrations: stereopure ionizable lipids at 10 mg/mL, cholesterol or DC-cholesterol at 5 mg/mL, C18PEG2K at 5 mg/mL, helper lipid (DOPE or DOTAP) at 5 mg/mL, and nucleic acid (for screens, aVHH mRNA and DNA barcode in a 9/1 w/w ratio) at 1 mg/mL.

2. For the 2 formulation LNPIT and LNP28 there is no significant difference between protein expression from mRNA NanoLuc and aVHH by these formulations in FaDu-derived tumors. However, for the PDX models, LNPIT was much more effective, which the authors note, but for which they do not suggest a reason. Could this be due to the difference in ECM or organisation in patient-derived tumours as opposed to the FaDu xenografts? We had a similar hypothesis (that ECM could be playing a role). We therefore added this to the text:

...We then repeated the experiment with aVHH; once again, LNP²⁸ delivered mRNA less efficiently than LNP^{IT}, which transfected human PDX cancer cells as well as murine cells (**Fig. 3b, Supplementary Fig. 6b**). The extracellular matrix and tumor microenvironment (TME) can contribute to intratumoral retention of therapeutic agents^{41,42} and may interact differently with constituents of LNPs. Differences in microenvironment may provide one potential mechanism that helps explain the gene transfer performance of LNP²⁸ versus LNP^{IT}.

3. In the authors' recent study (Journal of Controlled Release 2023, 353, 270–277) the most effective C12-200 lipid for a Cre- encoding mRNA in Ai14 mice (engineered with a CAG-Lox-Stop-Lox tdTomato construct) was one in which an S-epoxide was used. In the prior case a ~ 3-fold higher expression for the S-LNP was obtained over the R-isomer. The authors attributed this to better tolerability of the S-lipid in the mice rather than due to other structural parameters such as particle size, charge, stability. For this manuscript, the R-lipid was much more effective – why is this? We do not have enough data to make a definitive statement, but our leading hypothesis is that the route of administration could matter. We added this to the discussion.

[...] When the two lead LNPs were tested individually, they both delivered mRNA in FaDu tumors as predicted. Yet when the LNPs were tested in PDX tumors, only LNP^{IT} efficiently delivered mRNA. This second lesson is timely; in the past, testing several dozen LNPs in multiple *in vivo* tumor models would have required hundreds of animals. DNA barcoding makes this feasible with far fewer. When stereopure C12-200 isoforms were previously screened intravenously, the S-isomers yielded higher expression²⁷, while the most effective intratumoral LNP candidate here was composed of an R-isomer. This highlights the importance of screening via the intended route of administration when selecting LNP candidates for RNA therapeutics.

Reviewer 3 The mRNA vaccine has achieved great success and has also been attracted to be used in the treatment of various diseases, including the anti-tumor field. In this work, the authors developed the combinatorial application of mRNA and prodrugs to achieve antitumor responses. Fludarabine is a prodrug that does not have antitumor effects but can effectively produce a cytotoxic form after catalysis by purine nucleoside phosphorylase (PNP). The authors screened a series of LNPs that could specifically deliver PNP mRNA in tumor tissue but not other tissues, thus reducing the off-target toxicity of fludarabine. Overall, this is one of the attractive applications of mRNA technology, and most results support the conclusions well. Some comments are listed below.

1. In Figure 1c, C12-200 LNP showed the most efficient effect on Luc mRNA delivery after i.t. injection. It is recommended to use this original C12-200 LNP as a positive control to compare with LNP28 and LNPIT for i.t. delivery of mRNA. We performed this experiment and found that both LNPs outperformed C12-200.

...When compared with the LNP composed of racemic C12-200, both LNP²⁸ and LNP^{IT} yielded higher transfection in FaDu cells (**Supplementary Fig. 5b**).

Supplementary Fig. 5 | Characterization and efficacy of IT LNPs delivering NanoLuc- and aVHH-encoding mRNA. a, Diameter (nm), polydispersity index, pKa, zeta potential (mV), and encapsulation efficiency (%) on LNP²⁸ and LNP^{IT}. **b**, Both LNP²⁸ and LNP^{IT} had higher transfection of FaDu cells than the LNP containing racemic C12-200 (LNP^{C12-200})...

2. PNP is a foreign protein produced from mRNA. Will the injection trigger any unexpected immune responses? Our data suggest that LNP^{IT} encoding PNP did not lead to unexpected immune responses (e.g., tumor studies with LNP^{IT}-PNP but no drug afterwards did not affect tumor growth). However, we think this is an important point and therefore added it to the discussion.

... Finally, the expression of PNP, which is a bacterial protein, could elicit an unexpected immune response. While this will require further study, we did not find any evidence suggesting a broad, undesired immune response in our current experiments.

3. 2-Fluoroadenine is the ultimate antitumor effector. It is recommended to detect 2-fluoroadenine abundance in tumors when treated by PNP mRNA and fludarabine combination. We designed an *in vitro* assay to detect a 2-fluoroadenine analogue (MeP) for all six HNSCC models (two traditional cell lines and four PDX). We detected conversion from prodrug to activated drug and added these data as **Supplementary Fig. 12b**.

To evaluate therapeutic response in distinct human cancer cells, we also performed an *in vitro* cell killing assay (Fig. 6c). We plated six different HNSCC cell lines: human FaDu, murine MOC1, and four HNSCC patient-derived xenograft lines, termed PDX1, PDX2, PDX3, and PDX4 (Supplementary Fig. 12a). We treated cells with LNP^{IT}-PNP and MeP-dR (a nucleoside analogue of fludarabine that has served as a prototype for *in vitro* testing). We showed the expected PNP and nucleoside dependent cell killing and also observed time-dependent conversion from prodrug to oncolytic drug (Supplementary Fig. 12b). The negative controls, LNP-PNP^{IT} alone, MeP-dR alone, and no treatment behaved as expected. In contrast, cells treated with the positive control MeP (the toxic base released following MeP-dR hydrolysis by PNP) led to the expected ablation of cells in culture.

Supplementary Fig. 12 | PDX models and prodrug kinetics. ... b, Conversion kinetics of fludarabine into the activated oncolytic form by LNP^{IT}-PNP in all HNSCC models studied.

The corresponding Methods section for this assay was also added, as shown below:

***In vitro* PNP Activity Measurement.** MOC1, FADU or PDX cells were seeded into 6-well plates, and LNP^{IT} carrying PNP mRNA at 4.5 µg/well was added 24 h after plating. The following day, cells were incubated for 4 hours with 100 µM MeP-dR (substrate for *E. coli* PNP, see above) and the formation of MeP (cytotoxin) was measured using reverse phase HPLC (PerkinElmer).

4. According to point 3, it is recommended to add 2-fluoroadenine group as a positive control for anti-tumor response. We agree and added this positive control to our cell killing studies (please see Reviewer 1 Query 1).

To evaluate therapeutic response in distinct human cancer cells, we also performed an *in vitro* cell killing assay (Fig. 6c). We plated six different HNSCC cell lines: human FaDu, murine MOC1, and four HNSCC patient-derived xenograft lines, termed PDX1, PDX2, PDX3, and PDX4 (Supplementary Fig. 12a). We treated cells with LNP^{IT}-PNP and MeP-dR (a nucleoside analogue of fludarabine that has served as a prototype for *in vitro* testing). We showed the expected PNP and nucleoside dependent cell killing and also observed time-dependent conversion from prodrug to oncolytic drug (Supplementary Fig. 12b). The negative controls, LNP-PNP^{IT} alone, MeP-dR alone, and no treatment behaved as expected. In contrast, cells treated with the positive control MeP (the toxic base released following MeP-dR hydrolysis by PNP) led to the expected

ablation of cells in culture.

Figure 6 | LNP^{IT}-PNP and fludarabine phosphate combination therapy has anti-tumor effects validated across various HNSCC preclinical models. ... c, An *in vitro* assay to assess the anti-tumor cell effects of combination therapy in the syngeneic MOC1 murine line, FaDu human tumor cells, and four HNSCC patient-derived xenograft lines (PDX1, PDX2, PDX3, and PDX4). MeP-dR is an analogue of fludarabine phosphate (prodrug), used as a prototype compound for showing PNP activity. MeP is the toxic cleavage product of MeP-dR following PNP treatment, and serves as a positive control.

Reviewer 4 Summary: In this manuscript, the authors demonstrate that lipid nanoparticles (LNPs) are an efficient mRNA delivery system for the treatment of solid tumors. To design an optimized LNP for intratumoral delivery *in vivo*, they first used a pooled LNP library with a DNA barcoding system for multiplexing and identified a lead LNP, named LNP^{IT}, with enhanced transfection efficiency and minimal off-targets in both HNSCC xenograft and PDX mouse models. Using scRNA-seq analysis, they showed that cancer cells transfected with LNP^{IT} exhibited altered RNA and protein processing. Finally, by combining their LNP delivery system with mRNA encoding purine nucleoside phosphorylase, which converts fludarabine to 2-fluoroadenine (cytotoxic drug), they showed that LNP^{IT}-mRNA treatment leads to anti-tumor responses.

The present manuscript is well written and nicely demonstrates that the LNP-mRNA delivery system is an effective approach for the treatment of solid tumors. However, there are several important points that should be addressed to substantiate the authors' claims.

Major points:

1. Figure 4: The corresponding scRNA-seq analysis seems superficial and requires a more in-depth analysis to convincingly support the main claims. First, it is not clear how the authors quantified aVHH protein expression (Line 208) in Figure 4c and d. Did they use CITE-seq? If Figure 4c and d show aVHH mRNA expression, then Line 208 should be corrected. **For Fig. 4c-e, aVHH expression levels indicate aVHH protein expression measured via CITE-seq. This has now been clarified in the main text.**

...We then examined the expression levels of canonical marker genes for aggressive human HNSCC (*KRT14*, *KRT17*, *KRT6A*, *KRT5*, *KRT19*, *KRT8*, *KRT16*, *KRT18*, *KRT6B*, *KRT15*, *KRT6C*, *KRTCAP3*, *EPCAM*, *SFN*)⁴³ and found that this 12-gene signature was expressed primarily by cells in clusters 2 and 10 (**Fig. 4b**). Interestingly, we found different levels of LNP^{IT} aVHH delivery **quantified via cellular indexing of transcriptomes and epitopes by sequencing⁴⁴,**

with the highest amount in clusters 9 and 10 (**Fig. 4c,d, Supplementary Fig. 7a**). [...]

Additional experimental details were also added to the scRNA-seq Methods section clarifying this further:

...followed by tagging with TotalSeq™ anti-human Hashtag antibody (5 µg/mL; BioLegend) and oligo-tagged anti-VHH antibody (5 µg/mL). For the preparation of oligo-tagged anti-VHH antibody, 5' DBCO-modified oligonucleotide (CCTTGGCACCCGAGAATTCCAAAGTATGCCCTACGABAAA**A***A*, where * indicates phosphothioate bonds; GenScript) was conjugated to azide-modified rabbit anti-camelid VHH antibody (clone 96A3F5; GenScript) by click chemistry (GenScript)...

Second, the condition information of each cell (LNP^{IT}-mRNA or PBS control) should be displayed. As a negative control, the absence of aVHH mRNA expression in the PBS control should be shown. We agree and added this to the new **Supplementary Fig. 7a** (Note: this figure also includes data and analyses for other queries below):

Supplementary Fig. 7 | scRNA-seq analysis of PDX tumors treated with LNP^{IT} or PBS. **a**, aVHH protein expression quantified via CITE-seq in PDX tumors treated with LNP^{IT} or PBS. **b**, Complete heatmap of genes expressed across all clusters in PDX tumors treated with LNP^{IT} or PBS. Expression levels of **c**, stromal genes, **d**, CD47, **e**, SLA2, and **f**, PTPRC (or CD45) across all clusters in the PDX tumors.

Third, the authors should analyze cellular and molecular alterations induced by LNP^{IT}-mRNA transfection in all cell types including stromal and immune cells. We performed additional analyses of the clusters through HNSCC gene signatures for stromal cells, PTPRC/CD45 for immune cells, and SLA2 for immune cell infiltration in HNSCC (Supplementary Fig. 7c-f above).

We noted a number of variably up- or down-regulated genes in each cluster (Supplementary Fig. 7b). For example, we found clusters 2, 5, 8, and 10 had the highest gene expression of HNSCC stromal cell gene markers (ALDH1A1, BCL11B, BMI1, CD44)⁴⁶ (Supplementary Fig. 7c). We also found CD47 expressed ubiquitously across all clusters (Supplementary Fig. 7d). When evaluating SLA2, a prognostic marker in HNSCC that correlates with immune cell infiltration of the TME⁴⁷, we observed no expression (Supplementary Fig. 7e). This is consistent with low immune cell infiltration in an immunocompromised murine model.

Fourth, how do the authors distinguish aVHH+ from aVHH- cells? This was identified using anti-aVHH antibodies and CITE-seq. If there aren't aVHH+ cells in the PBS control due to ambient RNAs, the same DEG and pathway analysis should be performed in the PBS condition as a negative control. We added the DEG and pathway analysis, comparing the PDX tumors transfected with LNP^{IT} against the PBS-treated PDX tumors now as part of Supplementary Fig. 8:

a

Top 100 Differential Genes, LNP^{IT} vs PBS:

Gene	Log2FC	Log10FDR	FDR	Pvalue	Gene	Log2FC	Log10FDR	FDR	Pvalue
CHMP4B	0.822283	19.291214	5.1143E-20	0.49777	PTMS	0.416178	3.955736557	0.000111	0.337746
ZNFK1	1.506194	18.7885208	1.62734E-19	0.039738	SLA	-0.693617	3.955736557	0.000111	0.989769
STAT1	1.30299	16.2364611	5.80148E-17	0.014489	TRF51	-0.739744	3.955736557	0.000111	0.781821
TMSB10	0.927599	15.5763715	2.3833E-16	0.319897	CACPF2	0.403804	3.91413806	0.000122	0.218553
H3-3B	0.585163	14.5198307	3.02113E-15	0.976399	TMSB10P1	0.853336	3.91413806	0.000122	0.172029
PSME1	0.847556	14.4689927	3.39631E-15	0.832984	AKT3	0.815642	3.89073797	0.000129	0.466462
SOC31	1.999457	12.1687543	6.78023E-13	0.993818	HSP90AB3I	0.504857	3.879216104	0.000132	0.0020257
TDRD7	1.05844	11.6614253	2.18029E-12	0.120919	LVHE	1.372829	3.779765161	0.000166	0.008553
SLC25A22	1.146281	10.3064819	4.93762E-11	0.485294	DCP2	0.772958	3.773641089	0.000168	0.0019
FNDC3A	1.013676	9.83639981	1.45751E-10	0.038824	TMEM184E	0.625972	3.73464878	0.000184	0.342161
MAK	0.828013	8.89363746	1.2774E-09	0.976026	STO0A6	0.486774	3.714410382	0.000193	9.34E-05
PSME2	0.84443	8.491503	2.0348E-09	0.327362	UBE2C	0.504857	3.6994651	0.000246	0.037613
PEU1	0.764411	7.67312509	2.12263E-08	0.033371	UBR4	0.856249	3.64285896	0.000227	0.639064
ASCC3	0.98924	7.51564294	3.0504E-08	0.039213	MIER3	0.950823	3.6094651	0.000246	0.037613
ZUPT	1.311224	7.51550712	3.05136E-08	0.056646	DYNLL1	0.476072	3.527507311	0.000297	0.129536
AIDA	1.051466	7.32523322	4.72897E-08	0.804246	GDJ2	-0.367051	3.474187492	0.000336	0.777578
USP25	0.966008	7.0984679	7.97135E-08	0.84E-05	FRMD4A	0.601216	3.426208999	0.000375	0.446144
ETNK1	0.864615	7.06610009	8.58876E-08	0.003589	PSME2P1	0.832697	3.426208999	0.000375	0.064907
NAO2P	0.804168	6.8901153	1.27347E-07	0.92837	PABPC1	-0.34016	3.379311225	0.000418	0.051196
CND2	1.106129	6.80796075	1.55611E-07	0.731718	SOS1	0.750502	3.379311225	0.000418	0.069248
SNK2	0.773032	6.67598197	2.10872E-07	0.102873	FAM184B	1.087239	3.368622416	0.000428	0.859782
TRAF3	1.299721	6.35381137	4.42781E-07	0.629566	NAMPTP1	0.939805	3.346527286	0.000456	0.231168
VCPH1	0.747314	6.13849794	7.27043E-07	0.732327	GNB1	0.352428	3.254401655	0.000557	0.932255
CALM1	0.473976	5.85981549	1.38097E-06	0.057524	UBB	0.344362	3.217619467	0.000606	0.000265
MX1	1.133944	5.76062094	1.7352E-06	2.14E-05	AFTPH	0.559072	3.202469967	0.000627	0.168489
TOXNMT0	0.706629	5.7153078	1.92362E-06	0.160465	ACD2206	0.825497	3.202469967	0.000627	0.678397
MBD2	0.737325	5.45201715	3.53125E-06	0.137834	N4BP1	0.634818	3.202469967	0.000627	0.012525
ILRN	0.934704	5.36687469	4.2966E-06	0.09961	NAMPT	0.90532	3.197101508	0.000635	0.399811
MT-ND4	0.94813	5.17979426	6.1007E-06	0.226589	TENT5A	0.676016	3.118490569	0.000761	0.098853
MT-ATP6	0.953298	5.10805032	7.8991E-06	0.385697	BNMNSA	-0.760864	3.112195569	0.000772	0.003099
SIRPA	-0.740589	4.93539742	1.16039E-05	0.314947	MT-ND5	0.662869	3.104147204	0.000787	0.968359
LAMP2	0.489892	4.87140975	1.34459E-05	0.007881	UBE2L3	0.344036	2.9436448	0.001139	0.377115
MT-CO3	1.00399	4.86299919	1.37117E-05	0.063641	AC246787	-0.355807	2.9436448	0.001139	0.054399
AHNAK	0.53855	4.82416136	1.49913E-05	0.927073	POLR2F	-0.653888	2.906312729	0.001241	0.281564
RNF213	1.141034	4.72719719	1.87414E-05	0.640493	ELF1	0.469078	2.837982448	0.001452	0.071895
RNF130	-0.55048	4.71295432	1.93663E-05	0.794966	MT-ND4L	0.787862	2.749833091	0.001779	0.546259
TGFBRI	-1.051571	4.70127128	1.98948E-05	0.003339	EHRP1AS1	0.268847	2.690709359	0.002038	0.262817
NSD3	0.460904	4.66987036	2.1386E-05	0.021413	SKAP2	-0.694725	2.662099814	0.002177	2.17E-05
MT-ND3	1.000271	4.63858034	2.29837E-05	0.388913	VPS54	0.648296	2.662099814	0.002177	0.005171
XRN2	0.459442	4.61947254	2.40208E-05	0.001504	SLC33A1	0.766473	2.662099814	0.002177	0.152284
GREM1	0.860267	4.57483567	2.66149E-05	0.063911	MTND1P2	0.9432	2.662099814	0.002177	0.818513
IFIH1	1.290157	4.52324364	2.99748E-05	0.026491	AC103343	-0.368631	2.662099814	0.002177	0.497574
MT-CYB	0.952206	4.45156554	3.5537E-05	0.028143	ACTG1	0.239854	2.662099814	0.002177	2.3E-07
MT-ND2	0.355559	4.4304105	3.71184E-05	0.059923	MT-RNR1	0.333008	2.662099814	0.002177	0.380614
KLF6	0.632709	4.4304105	3.71184E-05	0.304795	RPL17P36	-0.319598	2.662099814	0.002177	3.42E-10
MT-ND1	0.852392	4.42638679	3.74639E-05	0.001929	GRN	0.860926	2.658702825	0.002194	0.993697
RAB3C	0.494528	4.28137251	5.23152E-05	0.01898	G3BP2	0.352467	2.626449057	0.002363	0.299918
ST00A10	0.57156	4.23938444	5.75726E-05	0.320891	ZNF800	0.56547	2.603262904	0.002491	0.616123
TMBIM6	0.384327	4.22243813	5.99186E-05	0.674859	HMGN3	0.672507	2.597229074	0.002528	0.749552
MT-CO2	1.010467	4.1689051	6.7779E-05	4.22E-05	IGF2R	0.994482	2.552540433	0.002802	0.629876

Full Reactome Pathway Analysis, LNP^{IT} vs PBS ($p < 0.01$):

b

Reactome pathway related to mRNA translation into protein

1. Translation
2. SRP Dependent co-translational protein targeting to membrane
3. Eukaryotic translation termination
4. Eukaryotic translation initiation
5. L13a-mediated translational silencing of ceruloplasmin expression
6. Cap-dependent translation initiation
7. Ribosomal scanning and start codon recognition
8. GTP hydrolysis and joining of 60S ribosomal subunit
9. Formation of ternary complex, and subsequently, the 43S complex
10. Formation of a pool of free 40S subunit
11. Activation of the mRNA upon binding of the cap-binding complex and eLFs and subsequent binding to 43S
12. Translation initiation complex formation
13. Eukaryotic translation elongation
14. Peptide chain elongation

Reactome pathway related to other protein metabolic processes

- | | |
|---|---|
| 15. Amyloid fiber formation | 22. Ovarian tumor domain proteases |
| 16. Post-translational protein modification | 23. Metalloprotease DUBs |
| 17. Protein ubiquitination | 24. Josephin domain DUBs |
| 18. E3 ubiquitin ligases ubiquitinate target proteins | 25. Asparagine N-linked glycosylation |
| 19. Synthesis of active ubiquitin: roles of E1 and E2 enzymes | 26. N-glycan trimming in the ER and Calnexin/Calreticulin cycle |
| 20. Deubiquitination | 27. Calnexin/Calreticulin cycle |
| 21. UCH proteinases | 28. ER Quality Control Compartment (ERQC) |

c

PDX2 transfected with LNP^{IT}

Supplementary Fig. 8 | Reactome pathway analysis for scRNA-seq data from LNP^{IT}. **a**, Differential gene expression of PDX tumors transfected with LNP^{IT} compared against PBS-treated PDX tumors, including the top 100 differential gene scores and full Reactome pathway analysis. **b**, Out of the 28 pathways upregulated by LNP^{IT} delivering aVHH mRNA to PDX HNSCC tumors, identified from the most upregulated genes, 14 of them are associated with mRNA translation into protein while the other 14 are related to other protein metabolic processes. **c**, Some of the top up-regulated genes identified *in vivo* in the PDX tumors transfected by LNP^{IT} were validated *in vitro* in a second PDX line, PDX2, via qRT-PCR.

A more detailed method for scRNA-seq data analysis including DEG and pathway analysis should be provided. We added details to the methods and are including an .xls with the differential gene expression analysis.

Fifth, the authors should characterize the molecular features of the cancer cell clusters in more detail. We agree and have included the full gene list above and performed additional

analyses on the clusters to understand their molecular features (**Supplementary Fig. 7c-f, also addressed in the response to your third point above**).

Finally, some up- and down-regulated genes in Figure 4e should be experimentally validated. We validated some of the most upregulated genes identified in the PDX tumors transfected *in vivo*. These were validated in a second PDX model (PDX2) transfected with LNP^{IT} *in vitro*, and the results are now mentioned in the text and included in the additional **Supplementary Fig. 8c** (see fourth query above). Added text:

Of these, 14 were directly associated with mRNA translation into protein (R-HSA-72766) and stemmed directly from the core of the metabolic pathway tree (Biological Process GO:0019538), consistent with previously reported transcriptional responses to mRNA-carrying LNPs³⁶ (**Fig. 4f,g, Supplementary Fig. 8b**). Some of the most upregulated genes found in the PDX model *in vivo* were also validated in a different PDX model (PDX2) transfected with LNP^{IT} (**Supplementary Fig. 8c**). These data provide an early line of evidence that HNSCC cancer cells respond to LNPs carrying mRNA in part by altering genes related to the manufacture and processing of RNA and protein.

Minor points:

1. Define “LNP” and “FaDu” in the abstract. “Lipid nanoparticles (LNPs)” is now defined in the abstract. FaDu is not an acronym but the actual name of that human cell line as defined in the ATCC website, so we left this as is.